# Dr-CiK: A Testbed for Foresight-Driven Agents

**Yihong Tang** [1 2]   **Andrew Robert Williams** [2 3 4]   **Arjun Ashok** [2 3 4]   **Vincent Zhihao Zheng** [1]   **Lijun Sun** [1]
**Alexandre Drouin** [2 5 4]   **Issam H. Laradji** [2 6]   **Étienne Marcotte** [2]   **Valentina Zantedeschi** [2 5]

## Abstract

Time series forecasting in real-world settings often depends not only on historical observations, but also on external context that must be actively discovered from noisy, heterogeneous information sources. Yet existing context-aided forecasting benchmarks typically assume that supporting context is already provided, leaving open whether agents can identify it on their own. When it comes to forecasting, Context is Key (CiK). However, identifying the right context from a large knowledge base and distilling it into forecast-useful evidence requires Deep Research (DR). Therefore, we introduce Dr-CiK, a benchmark for evaluating whether agents can retrieve forecasting-relevant supporting context from a document corpus, filter out distractors, distill the retrieved context into forecast-useful evidence, and generate forecasts supported by that evidence. Through context ablations and evaluations of state-of-the-art deep research and forecasting methods paired together, we show that high-quality context substantially improves forecasting performance in Dr-CiK. However, current DR agents recover only a small fraction of the ground-truth supporting evidence (usually $< 5\%$), are frequently misled by distractors ($> 80\%$ distractor citations), and can cause forecasters to perform worse with retrieved context than with no context at all. Our results motivate research on foresight-driven agents that search for the right context to predict the future. Dr-CiK is publicly available at https://servicenow.github.io/Dr-CiK.

## 1. Introduction

Time-series forecasters extrapolate trend and seasonality from historical observations (Hyndman & Athanasopoulos, 2018; Box et al., 2015). Recent deep-learning models (Lim & Zohren, 2021; Chen et al., 2023) and time-series foundation models (Rasul et al., 2023; Ansari et al., 2024; Woo et al., 2024) have widened applicability, but still largely operate on numerical history alone. In many real-world settings this is insufficient: predicting a demand spike during a heat wave, a ridership drop during a transit strike, or a disease shift after a policy change requires *external context* beyond the observed series.

A growing body of work studies *context-aided forecasting* (CAF), conditioning forecasts on textual context alongside numerical history (Xue & Salim, 2023; Gruver et al., 2024; Requeima et al., 2024; Williams et al., 2025). Yet existing CAF methods *assume* useful context is provided. In practice, relevant context must be discovered from heterogeneous, noisy sources. Recent *deep research* (DR) agents can search, synthesize, and cite multi-source evidence (Abaskohi et al., 2025), suggesting a path toward this harder setting. We study *context-aided forecasting via deep research* (CAF via DR): agents must identify predictive context, reject distractors, distill evidence, and generate evidence-grounded forecasts. Existing benchmarks evaluate retrieval or CAF in isolation but not both jointly (see Appendix A for a full survey).

We introduce **Dr-CiK**, a benchmark of 240 CAF via DR tasks with ground-truth supporting evidence and forecast targets. A key feature is that distractors are *forecast-dependent*: they match entity and domain, but lead to incorrect forecasts if used as context, requiring temporal and time-series reasoning to reject, not just surface-level relevance filtering. Dr-CiK's three-level evaluation protocol separates DR retrieval quality from forecasting quality, enabling failure attribution.

We find that DR quality causally drives forecasting outcomes: ground-truth supporting evidence reduces sCRPS nearly threefold over no context, yet current DR agents recover $< 5\%$ of supporting evidence and cite distractors in $> 80\%$ of cases, often making forecasts *worse* than no context. Our contributions are:

---
[1]McGill University [2]ServiceNow Research [3]Université de Montréal [4]Mila – Quebec AI Institute [5]Université Laval [6]University of British Columbia. Correspondence to: Yihong Tang <yihong.tang@servicenow.com>, Valentina Zantedeschi <valentina.zantedesch@servicenow.com>.

*Proceedings of the $2^{nd}$ ICML Workshop on Foundation Models for Structured Data*, Seoul, South Korea. 2026. Copyright 2026 by the author(s).

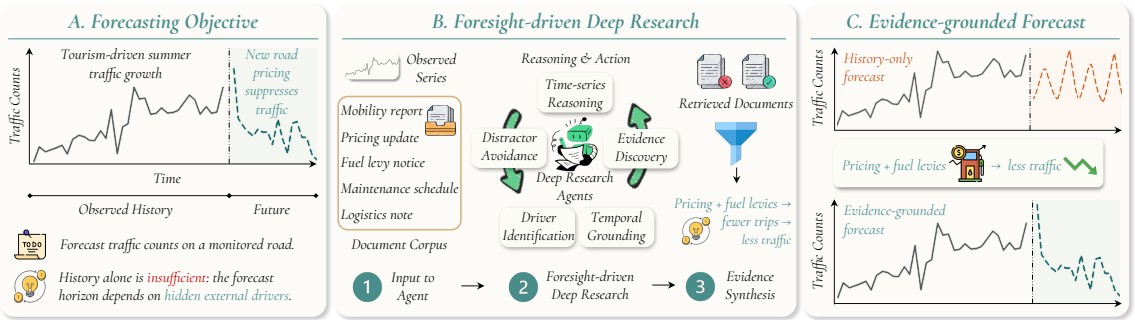

*Figure 1.* **CAF via Deep Research.** (A) History alone may be insufficient when future values depend on unobserved external drivers. (B) Given the series and a corpus, the agent ① receives the input, ② retrieves relevant context while avoiding distractors, and ③ synthesizes forecast-useful evidence. (C) A forecaster conditions on the evidence to produce a grounded forecast.

- We introduce the problem of CAF via DR and develop **Dr-CiK**, a benchmark of 240 tasks with ground-truth evidence enabling stage-wise failure analysis from retrieval to forecast.
- We define a three-level evaluation protocol isolating DR quality, distractor avoidance, and context utilization independently.
- We benchmark five DR agents paired with diverse forecasters and show supporting evidence improves sCRPS by nearly $3\times$ (see §I), while agents recover $< 5\%$ of evidence and cite distractors $> 80\%$ of the time, pointing to a wide gap for foresight-driven research.

## 2. Context-Aided Forecasting via Deep Research

Let $\mathbf{X}_H = [X_1, \ldots, X_t]$ be historical observations and $\mathbf{X}_F = [X_{t+1}, \ldots, X_T]$ the future. Traditional forecasting estimates $P(\mathbf{X}_F \mid \mathbf{X}_H)$; CAF conditions on text $\mathbf{C}$: $P(\mathbf{X}_F \mid \mathbf{X}_H, \mathbf{C})$ (Williams et al., 2025). In our setting, the agent receives $\mathbf{X}_H$ and accesses a corpus $\mathbf{D} = \{D_1, \ldots, D_n\}$. *Supporting documents* $\mathbf{D}_s \subset \mathbf{D}$ contain forecast-relevant content; *supporting evidence* $\mathbf{E}_s$ is the specific content to recover and synthesize. A good agent retrieves $\mathbf{D}_s$, extracts $\mathbf{E}_s$, and avoids distractors. Formally, conditioning on $\mathbf{E}_s$ should minimize forecast loss versus any other synthesized context $\mathbf{C}$:

$$\mathbb{E}_{\mathbf{x}_F} \mathcal{L}(P(\mathbf{X}_F | \mathbf{X}_H, \mathbf{E}_s), \mathbf{x}_F) \leq$$
$$\mathbb{E}_{\mathbf{x}_F} \mathcal{L}(P(\mathbf{X}_F | \mathbf{X}_H, \mathbf{C}), \mathbf{x}_F) \ \forall \mathbf{C} \in \mathcal{C}(\mathbf{D}). \quad (1)$$

## 3. Dr-CiK: A Testbed for Foresight-driven Agents

Dr-CiK is a testbed of 240 CAF via DR tasks and 8,849 documents (Figure 2), built around three pillars: (1) **Scale and Realism**: a scalable generation pipeline (applicable to any CAF dataset) releases 199 tasks from CiK (Williams et al., 2025) and GIFT-CTX for statistical robustness, plus 41 expert-annotated tasks on real, unedited series with domain-expert context; (2) **Controlled Difficulty**: multi-hop evidence decomposition with controllable depth, entity anonymization and time-shifting against memorization, and a five-class forecast-dependent distractor taxonomy; (3) **Fine-grained Evaluation**: ground truth for both DR and forecasting enables a three-level protocol, paired with difficulty annotations (§C).

### 3.1. Broad and Realistic Forecasting Scenarios

Tasks span six domains (energy, finance, transport, healthcare, retail, observability), frequencies from minute-level IoT to monthly macroeconomic, and histories of up to thousands of steps. Each task has a 1:2 supporting-to-distractor ratio (12 supporting documents, 25 distractors on average), ensuring surface-relevance retrieval surfaces mostly noise. The 199 generated tasks provide statistical breadth; the 41 expert-annotated tasks use real series and deployment-realistic context. Full split details are in Appendix B.

### 3.2. Task Generation Pipeline

Our scalable pipeline (Figure 6 in Appendix E) transforms multimodal time-series instances into DR environments in three phases.

**Phase 1: Entity Disambiguation.** Documents from all tasks are pooled into a single shared corpus, so entities may share names across tasks. We generate task-specific synthetic profiles, rewrite contexts accordingly, and shift time windows, mitigating lexical and numerical memorization.

**Phase 2: Multi-hop Decomposition.** Ground-truth evidence is decomposed into evidence units and expanded into an alternating evidence–reasoning chain with no branches or shortcuts. Hop count is the principal difficulty knob: longer chains require more documents and deeper synthesis (see §E for full details).

**Phase 3: Document Generation.** Supporting documents use an overlapping assignment strategy (consecutive doc-

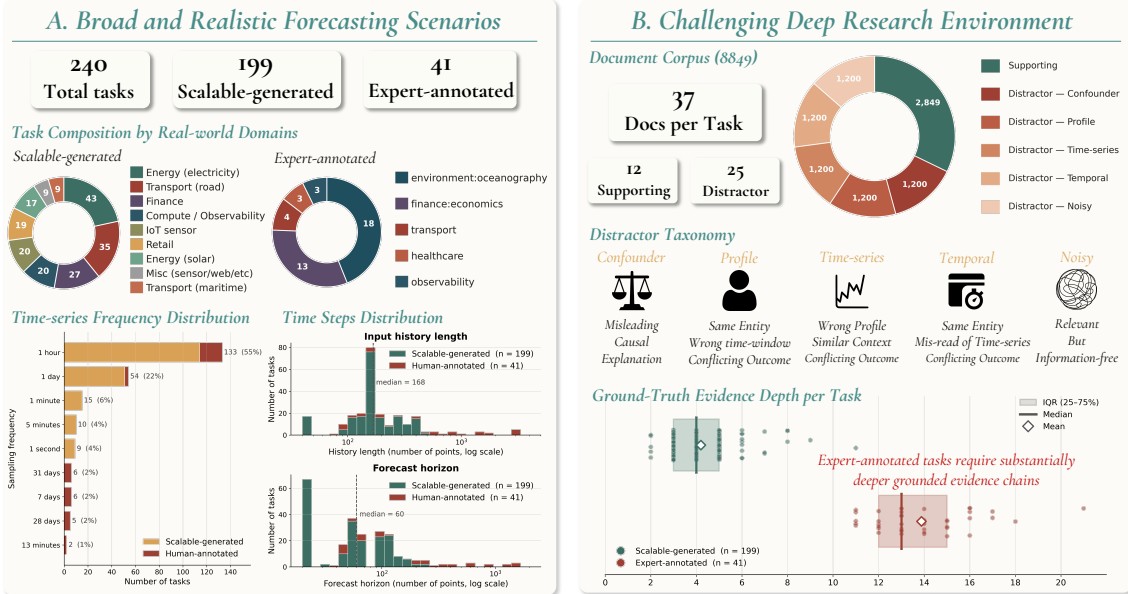

*Figure 2.* **Dr-CiK overview.** 240 tasks, 8,849 labeled documents, 5 distractor categories, a scalable generation pipeline, and a three-level evaluation protocol (§3).

uments share a causal anchor). Distractor documents are generated per our five-class taxonomy. A *generate-until-correct* protocol with human-calibrated LLM judges gates every stage; a final coding agent audits the completed environment with expert repairs (calibration details and all judge prompts in Appendix E.1).

### 3.3. Forecast-dependent Distractor Design

Dr-CiK introduces *forecast-dependent* distractors (Figure 2B): each distractor is generated from the ground-truth context and historical series to drive the forecaster toward a confidently wrong prediction, yet remains logically rejectable. We organize them into five classes: **Confounder** (spurious causal mechanism); **Profile Mismatch** (correct template, different entity, contradictory forecast); **Temporal Misalignment** (correct event, wrong time window); **Time-series Misinterpretation** (misreading of numerical history, stratified by scope $\times$ feature); **Noisy** (on-topic filler, no forecast-relevant content). This taxonomy probes causal, temporal, entity-level, and numerical reasoning simultaneously; examples are in Appendix F.

## 4. Experiments

**Three-level protocol.** Because Dr-CiK provides ground truth for both supporting evidence and forecasts, we can disentangle failure modes: **Level 1 (End-to-End)**: overall forecasting accuracy with DR-retrieved context; **Level 2 (Deep Research)**: retrieval quality via Evidence Recall, Sup. Doc. Recall, Distractor Avoidance; **Level 3 (CAF)**: forecasting under controlled evidence conditions.

| CONTEXT | Aurora | DP-Gemini | MoiraiAgent |
|---|---|---|---|
| NO CONTEXT | $0.48 \pm 0.06$ | $0.32 \pm 0.03$ | $0.34 \pm 0.03$ |
| ORIGINAL CONTEXT | $0.49 \pm 0.06$ | $0.23 \pm 0.03$ | $\underline{0.21 \pm 0.03}$ |
| BENCH2FUTURE | $\mathbf{0.48 \pm 0.06}$ | $0.63 \pm 0.07$ | $0.52 \pm 0.06$ |
| DRBENCH | $0.48 \pm 0.06$ | $0.57 \pm 0.06$ | $0.48 \pm 0.06$ |
| CODEX-GPT5.5 | $0.48 \pm 0.06$ | $\mathbf{0.33 \pm 0.03}$ | $\mathbf{0.31 \pm 0.04}$ |

*Table 1.* End-to-end sCRPS↓ ($\pm$ std. err.). Bold: best DR result per forecaster; underline: overall best.

**Setup.** We evaluate five DR agents (Codex (OpenAI, 2026), DRBench (Abaskohi et al., 2025), Bench2Future (Wildman et al., 2025), Open-Deep-Research (LangChain AI, 2025), Retrieval) paired with four forecaster families (statistical, pretrained, multimodal, zero-shot LLM with Direct Prompt (Williams et al., 2025), and a forecasting agent MoiraiAgent). Full model list in Appendix H; we report sCRPS (§I).

### 4.1. Level 1: End-to-End Results

Table 1 reveals a clear gap between oracle and DR-retrieved context. With original context, DP-Gemini and MoiraiAgent substantially outperform no-context, confirming they can leverage textual evidence. But Bench2Future and DRBench *degrade* performance below the no-context baseline, meaning their synthesized evidence is actively harmful. Codex is the only DR agent that stays competitive, and only with MoiraiAgent, suggesting a strong forecaster can mitigate DR noise but not compensate for weak evidence recovery. Aurora is essentially context-insensitive across all conditions (context definitions in §G).

**Distractor subtype distribution among cited distractors**

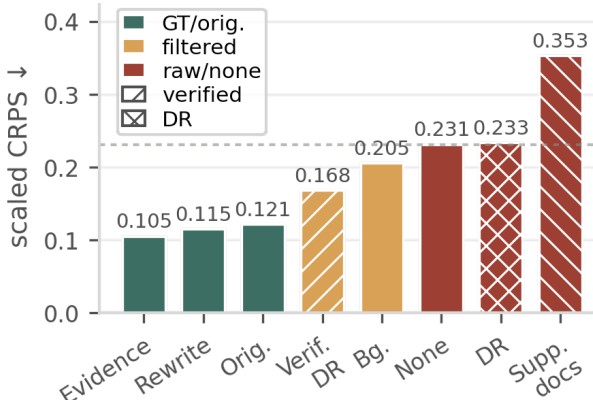

*Figure 3.* Distractor citation patterns across DR agents by type. Time-series distractors dominate retrieved noise despite equal corpus representation, showing agents cannot use the observed series as a filter on retrieved context (see §J for full analysis).

| DR | EVIDENCE RECALL | SUP. DOC. RECALL | DISTRACT. AVOID. |
|---|---|---|---|
| B2FUTURE | 3.8% | 7.5% | 22.7% |
| DRBENCH | 3.9% | 9.2% | 29.0% |
| RETRIEVAL | 4.3% | 10.0% | 20.4% |
| OPENDR | 4.8% | 9.9% | 23.5% |
| CODEX | **38.5%** | **48.9%** | **41.0%** |

*Table 2.* DR quality: Evidence Recall, Supporting Document Recall, Distractor Avoidance.

Two properties drive the gap: distractor presence and evidence coverage. Filtering agent output to remove distractors (Verified DR) consistently improves forecast accuracy (one-sided Wilcoxon test, significant), and supporting evidence coverage is positively rank-correlated with forecasting gains over no context (Figure 15, Appendix J).

### 4.2. Level 2: Deep Research Analysis

Evidence recall varies substantially across agents (Table 2): only Codex reaches 38.5%, while the rest hover near floor, showing that recovering ground-truth supporting evidence is non-trivial. Supporting document recall consistently exceeds evidence recall across all agents; agents find the right source documents more reliably than they extract the specific forecast-relevant content. Even Codex admits a majority of distractors (Distractor Avoidance: 41.0%), showing that rejection is as much a bottleneck as recall. Figure 3 shows that time-series distractors dominate retrieved noise despite being equally represented in the corpus: agents cite context plausible at the document level but inconsistent with the observed trajectory (§J), motivating agents that use the

series itself as a retrieval filter. Evidence recall degrades sharply as reasoning hop count increases, confirming multi-hop synthesis is the binding constraint (Appendix E). A qualitative failure-mode analysis (§K) further shows that even correct retrieval can fail: synthesis collapses numerical anchors and modal qualifiers in the supporting evidence.

### 4.3. Level 3: Context-Aided Forecasting

Under identical Codex-synthesized evidence, MoiraiAgent and DP-Qwen3.5 lead; scaling Qwen3.5 from 4B to 27B provides no improvement, suggesting model size is not the binding constraint (Appendix L). Figure 4 varies context conditions for DP-Gemini across eight conditions. Three findings stand out. First, oracle conditions (supporting evidence, original context, rewritten context) establish a ceiling at roughly half the no-context sCRPS; the small gap among them confirms Entity Disambiguation preserves forecast-relevant content. Second, raw Codex evidence is statistically indistinguishable from no context: the best DR agent provides *no measurable benefit* without verification. Third, raw supporting documents degrade performance *below* no context, confirming that an upstream synthesizer is necessary; forecasters cannot assemble evidence from raw documents on their own. Full results across all models and context conditions are in Appendix L.

## 5. Future Work

Supporting evidence reduces sCRPS nearly threefold, yet current DR agents recover evidence incompletely and are frequently misled by distractors, making synthesized context often *worse* than no context, highlighting the need for DR evaluation via downstream forecasting utility. Future work should extend coverage to more multimodal datasets (Zhang et al., 2025; Chang et al., 2025) and add numerical corpus entries, toward agents that integrate heterogeneous evidence for calibrated real-world forecasts.

*Figure 4.* sCRPS across context conditions for DP-Gemini. Lower is better.

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

# Appendix

**Table of Contents**

## A. Related Work

**Deep Research Agents and Benchmarks** DR benchmarks evaluate agents on multi-step search, retrieval, synthesis, and citation, treating retrieved reports as final outputs. ReAct (Yao et al., 2022) introduced the reason-and-act paradigm, while GAIA (Mialon et al., 2023) evaluates general-purpose agents on real-world QA. Recent large-scale DR benchmarks include DR Bench (Bosse et al., 2025), DeepResearch Bench (Du et al., 2025) (PhD-level reports), DRBench (Abaskohi et al., 2025) (enterprise DR), DRACO (Zhong et al., 2026) (factuality and citation quality), and DeepSearchQA (Gupta et al., 2026) (agentic retrieval). These benchmarks evaluate retrieved context or reports as final outputs, whereas Dr-CiK evaluates whether retrieved context improves downstream forecasting.

**Context-Aided Forecasting (CAF)** CAF studies how textual context improves time-series prediction. CiK (Williams et al., 2025) evaluates models on validated text–series pairs; TimeMMD (Liu et al., 2024) pairs series with textual reports; Aurora (Wu et al., 2026) and DoubleCast (Zheng et al., 2026) synthesize context for time series; TemporalBench (Weng et al., 2026) evaluates temporal reasoning under natural-language contexts. Reasoning failures in multimodal forecasting have been studied by Merrill et al. (2024) and Ashok et al. (2025). These works assume provided context; Dr-CiK tests whether agents can discover it from a noisy corpus.

**Deep Research Forecasting Agents** Some agentic forecasting benchmarks use downstream predictive performance as the target, but focus on categorical rather than numerical forecasting. ForecastBench (Karger et al., 2024) and ProphetArena (Yang et al., 2025) release live event-forecasting questions for real-time evaluation with web retrieval, but provide no controlled DR analysis. FutureX (Zeng et al., 2025) evaluates agentic reasoning for future prediction (categorical). Wang et al. (2024) integrates textual analysis into a multi-LLM forecasting pipeline with pre-filtered news. The closest work is Bench to the Future (BTF) (Wildman et al., 2025), a pastcasting benchmark with offline web snapshots, but its retrieval process is a black box and covers categorical rather than numerical time-series forecasting. Dr-CiK differs by enabling controlled analysis of how agentic DR affects numerical time-series forecasting accuracy, with labeled supporting evidence and forecast-dependent distractors that make it possible to evaluate not only *whether* an agent succeeds but *why*.

## B. Benchmark Composition

### B.1. Expert-Crafted Context-Aided Forecasting Tasks

The Context is Key benchmark (Williams et al., 2025) established context-aided forecasting as a measurable task, but several of its design choices limit its suitability as a target for retrieval. CiK tasks deviate from data an agent retrieving from real sources would encounter in at least one of the following ways: (1) they rely on synthetic time series or (2) on real series modified to reflect the supplied context, resulting in patterns that lack the richness of series produced by real-world processes, (3) contexts are not written by domain experts, hence do not necessarily reflect the register, terminology, or level of detail that a practitioner would write. We therefore curated a complementary set of expert-crafted tasks under different constraints: real, unedited series, context written by domain experts in their own register. This makes the expert-curated split of Dr-CiK suitable for evaluating whether agents can discover context in settings where the relationship between context and series is closer to real-world deployment.

#### B.1.1. TASK CREATION

| Domain | Tasks | Frequencies | History | Forecast |
|---|---|---|---|---|
| Finance & economics | 21 | B, D, W, ME, MS | 1,747 | 749 |
| Environment & oceanography | 20 | 1h, D | 1,503 | 645 |
| Transport | 17 | D, W, MS, QS | 107 | 47 |
| Healthcare | 13 | W, YS | 105 | 45 |
| Observability | 10 | 13min, 1h, 6h, 26h | 190 | 82 |
| All | 81 | — | 877 | 377 |

*Table 3.* Per-domain summary of the collected expert-annotated tasks. We report average history and forecast lengths in time steps.

The expert-curated split comprises 81 forecasting tasks built from unique time series drawn from seven publicly available sources, spanning five domains: environmental and oceanographic monitoring, finance and economics, healthcare, infrastructure observability, and transportation. (see Table 3 for a summary of characteristics). We then contracted SuperAnnotate

| Source | Tasks | Datasets |
|---|---|---|
| It's TIME (Qiao et al., 2026) | 54 | Coastal_T_S, Crypto, Global_Influenza, Global_Price, JOLTS, Job_Claims, Oil_Price, Port_Activity, SG_Carpark, SG_PM25, US_Term_Structure, Uncertainty_1M, Water_Quality_Darwin, azure2019, current_velocity, Finland_Traffic |
| BOOM (Cohen et al., 2025) | 9 | ds-141, ds-1082, ds-1324, ds-1396, ds-1424, ds-1440, ds-1501, ds-1761, ds-1849 |
| Eurostat | 7 | rail_pa_quartal, rail_go_quartal |
| OECD Health Statistics | 5 | DISCHARGE, STAY, BED_DAY |
| BTS TranStats | 4 | ATL, MSP, OH, SEA |
| Monash (Godahewa et al., 2021) | 2 | Melbourne_Pedestrian |

*Table 4.* Per-source summary of the collected expert-annotated tasks.

for crafting the corresponding contexts. A domain expert was commissioned to author a forecasting task by identifying meaningful patterns in the series and explaining their causes through structured causal chains. For that, the context creator was provided with a time-series plot, with both historical and future windows, and background information, a brief description of the quantity of interest. For each task, they were instructed to identify meaningful temporal patterns and compose two causal chains of events that could plausibly explain them: one explaining a pattern in the historical data window and one predicting a pattern in the future window (hence using future tense). Each causal chain contains one or multiple sequences of factors, from root causes to observable impacts, passing through intermediate effects.

In our experiments, we use 41 of these tasks and keep the remaining ones as a hidden test set for future evaluation. Because evaluating deep research agents requires calling proprietary models through external APIs, there is no guarantee that tasks sent for inference will not be retained, logged, or otherwise enter future training data. Withholding a portion of the expert-annotated set preserves a clean evaluation pool for re-testing once contamination of the public split becomes a concern.

**Pattern examples.** To anchor annotators on the kinds of patterns worth explaining, we provided a non-exhaustive list of nine pattern types: sustained regime change (a persistent shift to a new level or behavior), mean shift, slope or trend change, variance shift, seasonality change (in amplitude, phase, or shape), cycle-length change, spikes and drops (transient deviations), anomalies (isolated values incompatible with the surrounding behavior), and missing data segments. Annotators were free to identify patterns outside the list.

**From causal chains to benchmark tasks.** Each completed task consists of a time series, a description of the quantity of interest, and two causal chains: one historical and one forecast-oriented. The construction pipeline (Section E) transforms this material into the benchmark instance presented to agents: it derives supporting-evidence units from the causal chains, generates supporting documents that instantiate them, creates distractor documents that share surface features with supporting documents but are rejectable, and records difficulty annotations indicating which steps require temporal reasoning, time-series reasoning, or topical rejection.

### B.1.2. TIME SERIES STATISTICS AND PRE-PROCESSING

We source the expert-annotated time series from the datasets summarized in Table 4, with licenses and redistribution permissions listed in Table 5. A central design choice of the expert-annotated split is to select series whose forecast windows contain changes, such as anomalies, regime shifts, or other events that cannot be predicted from history alone. This ensures that each task has a non-trivial forecast that requires supporting context beyond the historical series. We also seek diversity across domain, sampling frequency, and sequence length, so that the expert-annotated split covers a broad range of real-world dynamics.

We preprocess series by resampling with mean aggregation to the target frequency and applying a 70/30 historical/forecast split. Two sources receive date adjustment as a contamination mitigation: Melbourne pedestrian counts are shifted forward by seven years and Eurostat rail by nine years, projecting their date ranges into a window not covered by current model training cutoffs.

| Source | License | Redist. | License reference |
|---|---|---|---|
| CiK | Apache 2.0 | Permitted | huggingface.co/datasets/ServiceNow/context-is-key |
| It's TIME | Apache 2.0 | Permitted | huggingface.co/datasets/Real-TSF/TIME |
| Datadog BOOM | Apache 2.0 | Permitted | huggingface.co/datasets/Datadog/BOOM |
| Eurostat | CC BY 4.0 | Permitted | ec.europa.eu/eurostat/help/copyright-notice |
| OECD Health Statistics | OECD Terms and Conditions | Permitted | oecd.org/en/about/terms-conditions.html |
| BTS TranStats | US Public Domain (17 U.S.C. §105) | Permitted | 17 U.S.C. §105 |
| Monash Repository | CC BY 4.0 | Permitted | huggingface.co/datasets/Monash-University/monash_tsf |

*Table 5.* Data sources, licenses, and redistribution permissions for the benchmark sources.

### B.2. Scalable Generated Split

The scalable generated split contains 199 tasks derived from CiK (Williams et al., 2025) and GIFT-CTX[1]. We construct this split by prioritizing task families that reflect the core reasoning demands of Dr-CiK: forecasting under context-driven distribution shifts such as event-induced demand changes, operational outages and maintenance, anomaly correction, holiday-conditioned traffic variation, and weather-state transitions. We exclude thin constraint-only items, overly verbose near-duplicate variants, and distributionally narrow reserve-style families that are less representative of the target task. To avoid overlap between sources, we also remove GIFT-CTX examples whose raw source field indicates CiK provenance, so that the GIFT-CTX portion provides complementary rather than duplicated coverage.

## C. Difficulty Annotations

To characterize the reasoning demands of Dr-CiK, we annotate each task along four difficulty dimensions: context depth, context explicitness, domain knowledge, and temporal complexity. These annotations are used for diagnostic analysis rather than filtering: they allow us to relate DR and forecasting failures to the kinds of reasoning required by each task.

Figure 5 shows that the expert-annotated split is substantially harder than the scalable generated split across several dimensions. In the scalable generated split, most supporting evidence is explicit, most tasks rely on general rather than specialist domain knowledge, and temporal structure is usually straightforward. By contrast, expert-annotated tasks contain more uncertainty, more implied evidence, more specialist knowledge, and more variable temporal structure. This confirms that the two splits play complementary roles: the scalable generated split provides broad coverage and statistical power, while the expert-annotated split stresses deployment-realistic reasoning where the relevant context is less direct and harder to operationalize for forecasting.

## D. Fine-Grained Diagnostic Dimensions

Beyond the difficulty arising from the number of supporting documents and the presence of distractors, we identify four further dimensions along which a task can be hard. Each captures a distinct source of textual reasoning difficulty (Figure 2C), and the dimensions are independently controllable, allowing aggregate accuracy to be decomposed into a per-dimension diagnostic profile and specific failure modes to be isolated.

**Domain Knowledge.** Whether a causal step can be traversed using common-sense world knowledge or requires specialized

---

[1]https://huggingface.co/datasets/Salesforce/GIFT-CTX

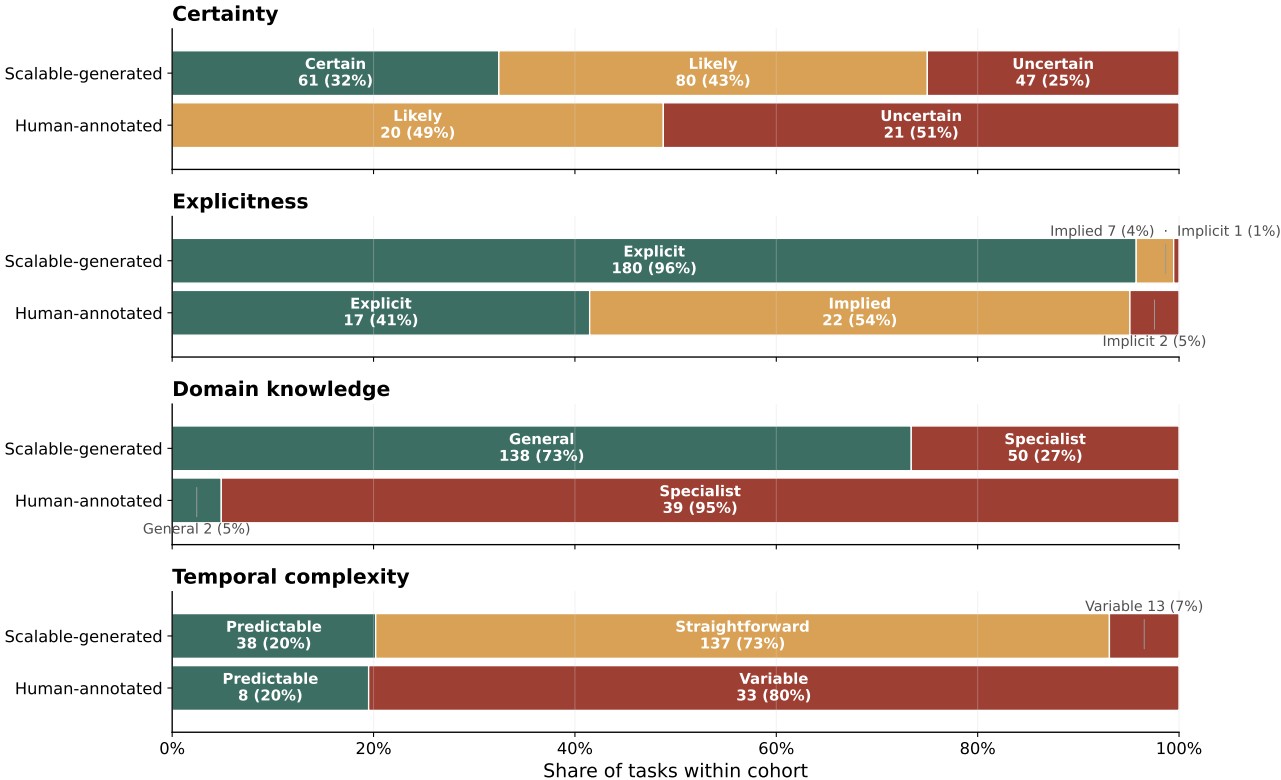

*Figure 5.* **Difficulty annotations for scalable generated and expert-annotated tasks.** Each row compares the distribution of a difficulty dimension across the scalable generated split and the expert-annotated split. Expert-annotated tasks are consistently harder: they contain a larger share of uncertain context, implied rather than explicit evidence, specialist domain knowledge, and variable temporal structure. In contrast, the scalable generated split provides broader coverage but is more often explicit, general-domain, and temporally straightforward.

expertise. We grade each causal edge as either *General* (e.g., *heat causes increased air conditioning use*) or *Specialist* (e.g., *water column stratification from freshwater input traps fine particles near the surface and amplifies turbidity readings*). A trigger may be correctly identified at the surface level, but the causal connection cannot be drawn without domain-specific knowledge.

**Temporal Complexity.** The richness of temporal reasoning needed to translate a causal step into its quantitative effect on the time series. We use three levels. *Straightforward* effects are immediate and bounded (a storm passes and turbidity spikes within hours). *Predictable* effects involve delayed onset, moderate duration, and recurring but regular patterns (the wet season arrives in November and turbidity shifts into an elevated regime over the following weeks). *Variable* effects are long-lagged, cumulative, or compound across irregular recurrences (years of sediment accumulation amplify each storm's peak turbidity over time). Higher levels require the agent to place triggers correctly relative to the forecast window and to attribute cumulative effects across multiple events.

**Explicitness.** How directly the relevant information is signaled within an individual document chunk. *Explicit* chunks name the quantity of interest, state the trigger, and quantify the effect (*turbidity at the Darwin buoy spiked to 162 NTU following Tropical Cyclone Marcus*). *Implied* chunks mention the trigger but leave the connection to the target qualitative or scope-shifted (*Cyclone Marcus generated 4–6m wave heights across the Timor Sea region*). *Implicit* chunks describe a condition with no explicit connection to the target, requiring the agent to construct the link by reasoning across geography, population, or context (*the 2017–18 wet season was declared one of the most active cyclone seasons in the northern Australian basin in a decade*). Lower explicitness penalizes agents that retrieve by surface similarity rather than causal reasoning.

**Certainty.** The epistemic status of the information itself, independent of how clearly it is signaled. *Certain* chunks

report directly observed values with negligible error (*measured 162 NTU at 14:15 on December 7th*). *Likely* chunks use probabilistic language or modeled estimates (*wave heights estimated at 3–5m, suggesting turbidity likely exceeded 100 NTU*). *Uncertain* chunks compound multiple sources of uncertainty and require explicit uncertainty reasoning (*projected increases in cyclone intensity could push peak turbidity beyond historical maxima next season*). Even correctly retrieved chunks carry uncertainty that must be propagated into the forecast rather than treated as ground truth.

**Coverage Across Difficulty Levels.** These dimensions also motivate Dr-CiK's dual-partition design. The scalable partition provides a broad, relatively balanced baseline across difficulty levels, while the expert-annotated partition is concentrated at the harder end: 95% of expert-annotated tasks contain at least one Specialist step, 80% involve Variable temporal effects, and none are supported by entirely Certain evidence. The two partitions together evaluate routine context use alongside the harder cases where agents must reason under uncertainty, domain specificity, and temporal complexity.

**Preliminary Observations.** Decomposing errors along our diagnostic axes exposes hidden failure modes. For *Certainty*, the strongest language-aware forecaster's sCRPS degrades from 0.112 on *Certain* cases to 0.191 on *Uncertain* cases, which is the largest gap across axes and suggests that these labels capture textual rather than temporal difficulty. For *Explicitness*, DR augmentation effects scale monotonically: it worsens sCRPS by 0.304 on *Explicit* cases and by 0.163 on *Implied* cases, but marginally improves *Implicit* cases. This taxonomy conditioning reframes DR's utility from a global average to a context-dependent property.

# E. Environment Generation Pipeline

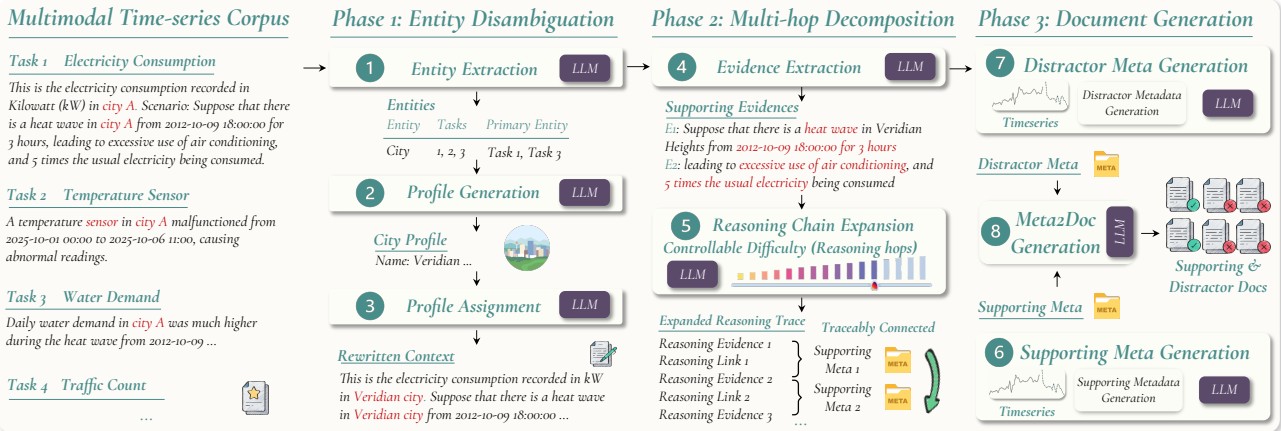

*Figure 6.* **DR Environment Generation Pipeline. (1) Entity Disambiguation (Steps 1–3):** Entities mapped to synthetic profiles to ensure identifiability and mitigate memorization. **(2) Multi-hop Decomposition (Steps 4–5):** Evidence fragmented into a multi-hop chain; hop count controls retrieval breadth. **(3) Document Generation (Steps 6–8):** Supporting and distractor documents synthesized from the chain and distractor taxonomy. Human-calibrated LLM judges enforce integrity at every stage.

## E.1. Human-Calibrated LLM Judges

Dr-CiK uses LLM judges to verify intermediate outputs during task generation, but we do not rely on them out of the box. Instead, we calibrate each judge against human review before using it in the full generation pipeline. The goal of calibration is conservative filtering: a selected LLM judge should agree with the human majority when possible, and otherwise be stricter rather than more permissive.

**Human calibration set.** We first generate a representative set of preliminary tasks and ask human reviewers to label the major generation stages: ground-truth supporting-evidence extraction, reasoning-chain expansion, supporting-document generation, and distractor-document generation. Across 12 calibration tasks and four generation stages, seven reviewers provided 117 human review records, comprising 79 stage-level judgments and 38 fine-grained per-entry judgments. Figure 9 shows the reviewer dashboard, and Figure 10 shows the detailed one-to-one review interface. For each stage, reviewers inspect the relevant inputs and outputs, then assign a binary judgment indicating whether the output is acceptable. When an output is marked incorrect, reviewers can provide free-form notes describing the failure mode.

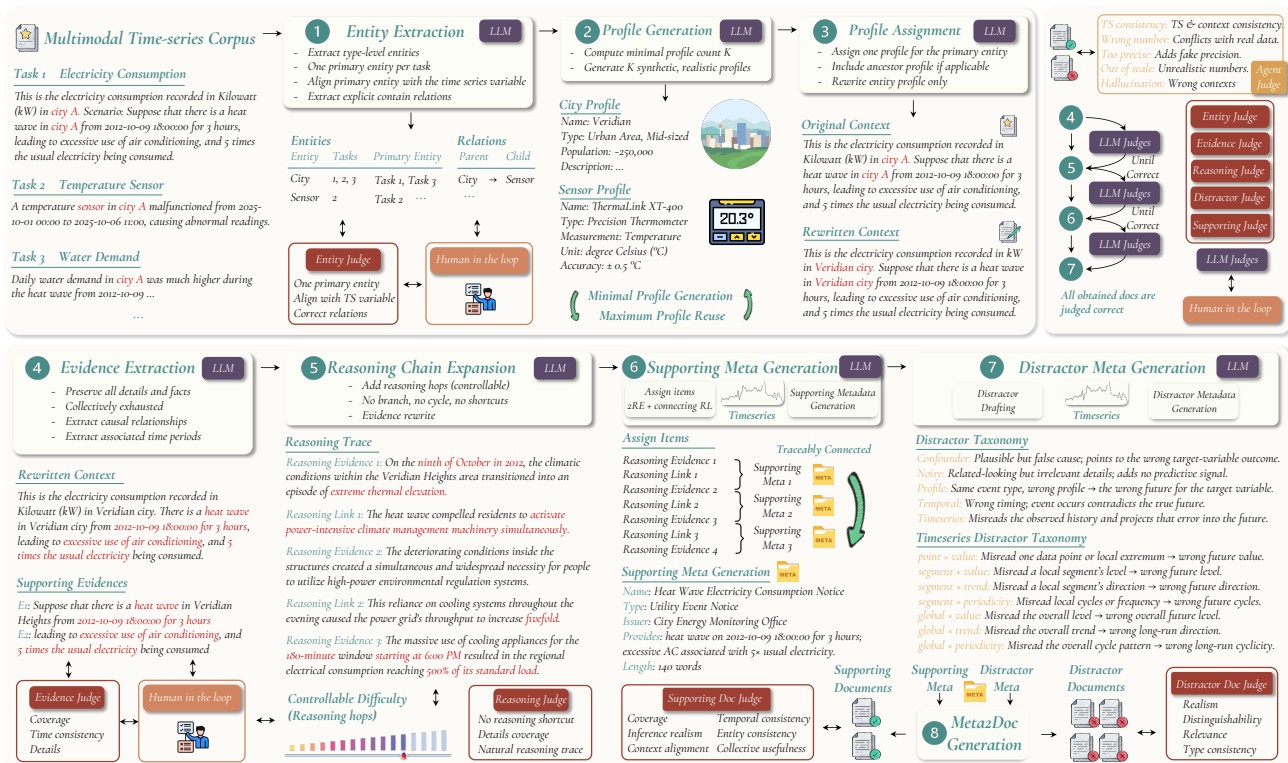

Figure 7. **Worked Example of the DR Environment Generation Pipeline.** This expanded view instantiates the pipeline in Figure 6 on a concrete CAF task. Starting from a multimodal time-series instance, the pipeline (**1**) extracts entities, assigns synthetic profiles, and rewrites the original context to reduce entity ambiguity and memorization shortcuts, (**2**) decomposes the rewritten supporting context into supporting-evidence units and expands them into a controllable multi-hop reasoning chain, and (**3**) converts the resulting reasoning trace into supporting documents while generating forecast-dependent distractors from the distractor taxonomy. The right-hand audit loop shows the local LLM judges and final Agent Judge used to verify intermediate outputs and repair residual inconsistencies with human oversight.

**Judge calibration.** For each generation stage, we design a specialized LLM judge with stage-specific acceptance criteria. These include an Evidence Judge for checking whether supporting-evidence units cover the rewritten context, a Reasoning Judge for validating reasoning-chain structure and completeness, a Supporting Document Judge for verifying that generated supporting documents preserve the intended evidence, and a Distractor Judge for checking realism, relevance, and rejectability of distractor documents. We iteratively refine each judge prompt until the selected LLM judge is at least as strict as the human majority on the calibration set. As shown in Figure 8, the final selected judges either agree with the human majority or reject more outputs than the human majority in nearly all comparable cases. Full judge prompts and rubrics are provided in § M.

**Regeneration after calibration.** The preliminary tasks used for calibration are not included in the final benchmark. After calibration, we discard those tasks and regenerate the benchmark from scratch. During this final synthesis run, each pipeline stage is guarded by its corresponding calibrated LLM judge. We use a *generate-until-correct* procedure: if an intermediate output fails its judge, that output is regenerated and rechecked until it passes. This procedure is applied separately to evidence extraction, reasoning expansion, supporting-document generation, and distractor-document generation. The prompts used for each generator and judge are listed in § M.

**Judge models.** After calibration, we use a fixed judge assignment for full-scale generation. Entity validation is performed by `google/gemini-3-flash-preview`[2]. Ground-truth evidence extraction and supporting-document validation use `openai/gpt-5.4-nano`[3], while reasoning-chain validation and distractor-document

---

[2] https://openrouter.ai/google/gemini-3-flash-preview
[3] https://openrouter.ai/openai/gpt-5.4-nano

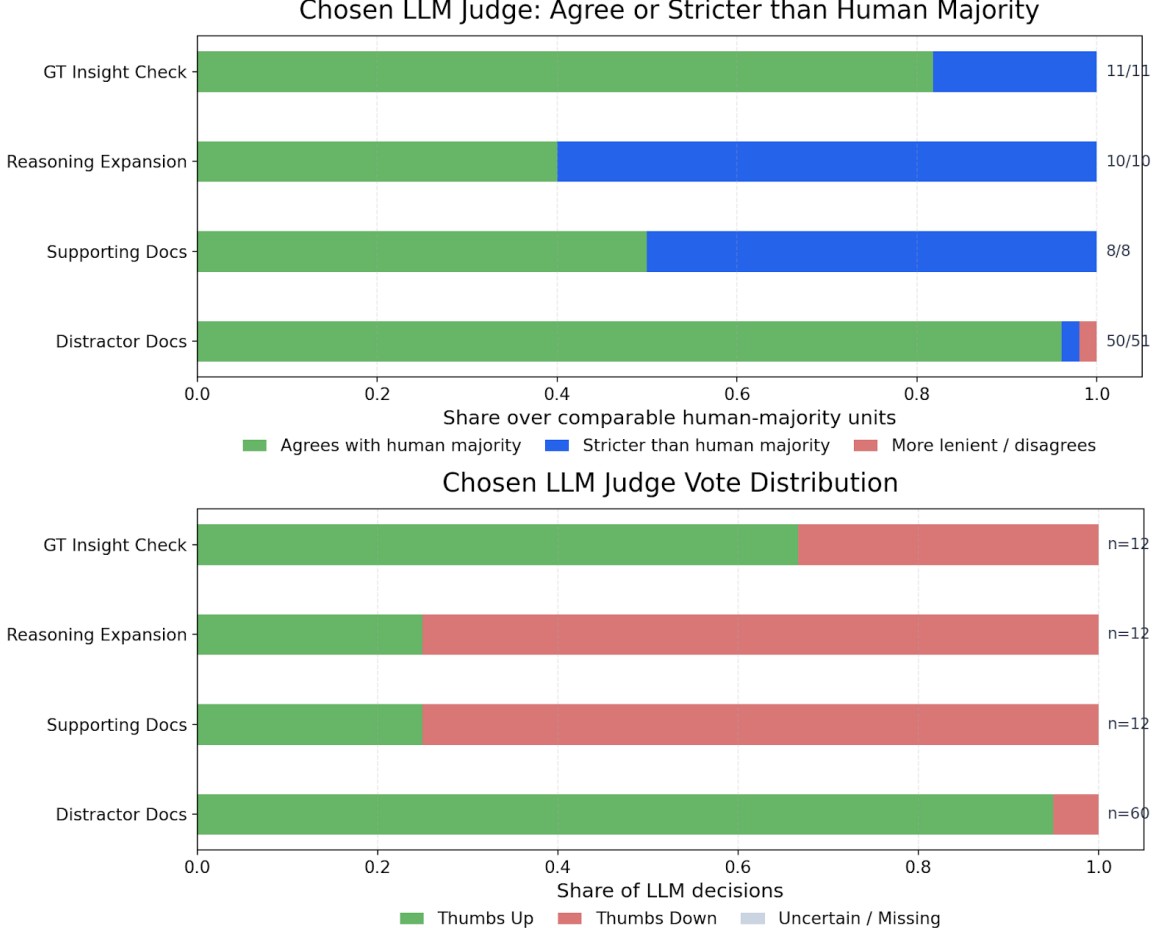

*Figure 8.* **Calibration of selected LLM judges against human majority labels.** For each generation stage, we compare the selected LLM judge against human majority decisions on a held-out calibration set. The top panel reports the share of comparable units where the chosen LLM judge agrees with the human majority, is stricter than the human majority, or is more lenient/disagrees. The bottom panel shows the final vote distribution of the chosen LLM judge. The selected judges are calibrated to be conservative: when they do not agree with the human majority, they are intended to reject rather than accept borderline outputs.

validation use `openai/gpt-5.4-mini`[4]. For downstream evaluation, evidence-recall matching is judged by `google/gemini-3-flash-preview`. The final global audit is not part of the automatic generation pipeline: we use Claude Code[5] as an agentic auditor to inspect the completed environment, flag cross-document inconsistencies or leakage, and guide targeted human repairs.

**Global agent audit.** Local judges verify individual stages, but some errors only become visible at the completed-environment level, such as inconsistencies across documents, leakage between supporting and distractor documents, or conflicts between the generated corpus and the time-series metadata. We therefore apply a final Agent Judge to audit the complete DR environment after all documents are generated. The Agent Judge checks global consistency, source-document alignment, distractor rejectability, and whether the final corpus supports the intended CAF via DR task. Flagged issues are repaired with human oversight. The Agent Judge prompt is also provided in § M.

---

[4] https://openrouter.ai/openai/gpt-5.4-mini
[5] https://www.anthropic.com/claude-code

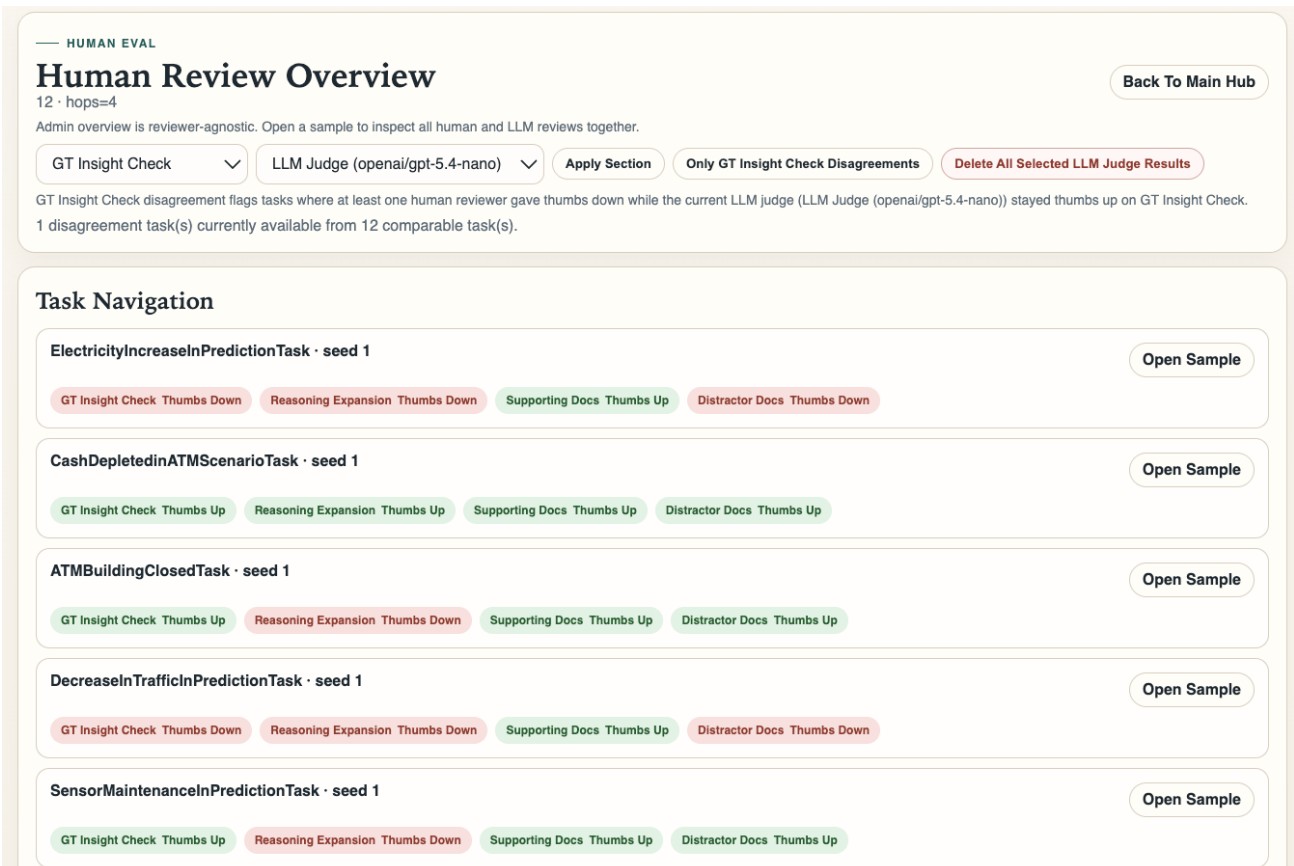

*Figure 9.* **Human review dashboard.** The dashboard summarizes human and LLM-judge decisions across sampled tasks and generation stages. Reviewers can filter by stage, inspect disagreement cases, and open individual samples for detailed comparison. This interface is used to collect human majority labels for calibrating the LLM judges before full-scale benchmark generation.

## F. Distractor Examples

For each example below, we display the time series and task background separately and reproduce here only the distractor document content, its type, and an analysis of why the document fits that type. The two source tasks (`task_A`: 4-week moving average of Initial Claims, `task_B`: Global Price of a Standard Industrial Raw Material) together cover all five distractor classes and every $scope \times feature$ time-series sub-combination that appears in the released bundle.

### F.1. Task A: 4-Week Moving Average of Initial Claims

**Background.** The Urban-Industrial Employment Ledger tracks the 4-week moving average of Initial Claims, sampled weekly from January 2022 to September 2025.

**Ground-truth Supporting Evidence.** The task carries two human-labeled causal chains, one explaining the history (HC1) and one specifying the projected future (HC2), both are anchored to a shared background event (E1). Distractor analyses below should be read against this ground truth.

---

**Task A: Initial Claims**

```
History mechanism

1. The central regulatory body's aggressive monetary tightening cycle, raising the base lending rate from
approximately 0.25% in early 2022 to approximately 5.3% by mid-2023, created competing pressures on the
Urban-Industrial Employment Ledger.
```

---

```
2. Post-pandemic retention strategies, where firms kept staff despite slowing demand to avoid rehiring costs
in a tight environment, suppressed initial claims filing rates below structural equilibrium levels. This
phenomenon, combined with historically low resignation rates in 2022, drove the 4-week moving average from
~240,000 toward the ~200,000 trough observed in early 2023, well below the pre-pandemic structural baseline
of ~220,000.
3. As these tight conditions in the Urban-Industrial Employment Ledger began to reverse under monetary
tightening, the rate hikes began transmitting into tighter financial conditions by mid-2023. Rate-sensitive
sectors, particularly technology, real estate, and financial services, initiated significant layoff cycles.
This sector-specific shock drove the sharp spike from ~200,000 to ~250,000 between January and July 2023,
while more resilient sectors like healthcare and government employment partially offset the broader
deterioration.
4. As these layoffs spread and financial conditions remained restrictive, the early-2024 reabsorption of
workers from tech and financial sectors into healthcare, logistics, and services only partly offset the
broader pass-through of restrictive credit conditions. As tighter financing and slower hiring spread beyond
the first wave of rate-sensitive layoffs, initial claims rose again toward the mid-230,000s by mid-2024
rather than stabilizing at the early-2024 trough.
5. The 4-week moving average of initial claims within the Urban-Industrial Employment Ledger fell from
roughly 240,000 in early 2022 to near 200,000 by early 2023, rebounded to about 250,000 by mid-2023 as
rate-sensitive layoffs intensified, eased back toward roughly 200,000-205,000 around early 2024, and then
climbed again toward roughly 235,000-240,000 by mid-2024 as restrictive financial conditions continued to
broaden across employers.

Forecast mechanism

1. Restrictive monetary policy continues to maintain elevated borrowing costs, with the lagged transmission
of high interest rates into business investment and hiring decisions still working through the economy.
2. The sustained inversion of the treasury yield curve through 2024 is projected to continue compressing net
interest margins for regional banks, tightening credit availability for small and medium enterprises.
3. Since these enterprises account for a significant part of the Urban-Industrial Employment Ledger,
credit-constrained hiring and episodic layoffs in this segment are expected to generate recurring upward
spikes.
4. Given this tightening in credit conditions, sectors that over-hired during the 2021-2022 expansion,
particularly technology, e-commerce, and professional services, are projected to continue rightsizing their
workforces, contributing to episodic layoff waves that can drive the claims above 240,000.
5. However, the rightsizing-driven layoffs remain sector-specific rather than economy-wide. Continued
strength in healthcare, government, and infrastructure hiring partially offsets these losses, keeping the
oscillations in the Urban-Industrial Employment Ledger from developing into a sustained uptrend, so that
spikes toward 240,000-246,000 remain temporary rather than establishing a new structural baseline.
6. The 4-week moving average of initial claims in the Urban-Industrial Employment Ledger is projected to
fluctuate within an elevated range through September 2025, with most weekly readings expected to fall
between 220,000-235,000, reaching a low near 213,000 and a high near 246,000 only at the extremes of the
oscillation, with a periodicity of approximately 6-8 weeks and no sustained directional trend.
```

**Confounder.**

> "Beginning in early 2022, the Central Regulatory Body transitioned its primary regulatory focus toward the integration and enforcement of carbon-neutral manufacturing standards. ... current surges in claims data are primarily influenced by the implementation of mandatory retraining programs for the green energy transition."

The document keeps the same entity, variable, and time window, but replaces the true causal mechanism, aggressive monetary tightening from ∼0.25% to ∼5.3%, with an entirely different causal chain, carbon-neutral mandates and mandatory retraining cycles. The proposed cause is plausible and on-topic but does not in fact drive the target series, which is the defining property of a confounder: a spurious causal explanation whose effect lands on a different mechanism than the one governing the future trajectory.

**Noisy.**

> "This manual establishes the protocols for the administration, data integrity, and systematic auditing of the Urban-Industrial Employment Ledger. ... Reporting must encompass all active personnel changes, including new hires, terminations, and shifts in industrial labor hours."

The document is realistic, on-topic at the surface, refers to the same ledger and same agency, and is full of detail, but its content is exclusively administrative: classification tiers, submission windows, and digital-transition protocols. None of it carries forecast-relevant information about claims behaviour. The document tests whether the agent anchors on realistic peripheral filler instead of separating supporting evidence from noise.

**Profile.**

"**Issuer:** Metropolitan Workforce Registry. ... Labor volatility ... is being fundamentally reshaped by a dual transition ... green energy sectors ... industrial automation. ... the 4-week moving average of initial claims remains atypically depressed, consistently tracking within a narrow range between 180,000 and 195,000."

The scenario template, labor ledger, 4-week moving average of initial claims, and contemporary timeframe, is preserved, but the entity is swapped to a different same-type registry, *Metropolitan Workforce Registry*, with different drivers, automation and skill-matching efficiency. The implied future is a tight 180k to 195k band, contradicting the realised future, which fluctuates in the 213k to 246k range. This is the canonical profile distractor: maximal scenario-level overlap, contradicting forecast.

**Temporal.**

"This analysis examines the specific interactions between monetary policy adjustments ... during the critical **2015 to 2016** window ... This tightening cycle moved the benchmark interest rates from a baseline of 0 percent to a peak of 0.75 percent."

The cause type, monetary tightening, and entity, the same ledger, are correct, but the entire event is anchored to 2015 to 2016, which lies before the historical observation window from 2022 to 2025. The rate path described, 0% to 0.75%, is also inconsistent with the 0.25% to 5.3% trajectory governing the actual forecast window. Same mechanism, wrong temporal anchor: the document has no predictive bearing on the prediction window.

**Time-series,** $global \times value$**.**

"A longitudinal review of the Urban-Industrial Employment Ledger reveals a distinct pattern of **diminishing peaks** ... future oscillations in the 4-week moving average of initial claims are expected to dampen significantly."

The document misreads the global level and extremum profile of the history. Empirically, peaks taken over rolling 8-week windows are roughly stable around 240k to 252k, not diminishing, with `240k` early, `251k` mid-history, and `236k` late. Propagating the false "dampening" assertion forward produces a forecast that is too tightly bracketed near the mean, missing the realised peaks up to 246k.

**Time-series,** $segment \times periodicity$**.**

"... a persistent departure from symmetric oscillation, favoring a distinct asymmetric profile ... a gradual build and a slow, incremental climb in filing rates ... filing rates do not gradually recede, instead, they are observed collapsing back to the floor immediately after a peak is reached."

The document misstates the periodic structure of local segments, replacing the actual symmetric multi-week oscillation, weekly autocorrelation $\approx 0.40$ at lag 7 and $0.46$ at lag 6, with an asymmetric "slow rise / vertical drop" sawtooth. An agent that internalises this segment-level periodicity description will project long upward runs ending in abrupt collapses, which does not match the realised mostly-symmetric fluctuations.

**Time-series,** $global \times trend$**.**

"... have formally entered a phase of **secular expansion** ... Each successive market trough ... is now being established at a progressively higher level than the previous one. ... projected to consistently break above the 250,000 threshold."

The document misreads the global trend, asserting a sustained ascending trajectory and a breakout above 250k. The actual history is range-bound: troughs over rolling 8-week windows do not monotonically rise, they vary between ∼197k and ∼236k, and the realised future stays within 213k to 246k, never breaking 250k.

**Time-series,** $global \times periodicity$**.**

"... characterized by high-momentum phases where directional shifts ... persist once established. ... the prevailing time series lacks short-term cyclicality or mid-year reversals ... projected to continue their ascent and break past the 260,000 level."

The document misreads the global periodicity by asserting that the series is non-oscillatory and lacks short-term cyclicality, and uses this to justify a monotonic forecast above 260k. This contradicts the observed periodicity, autocorrelation $\approx 0.63$ at lag 4 and $0.40$ at lag 7, and the realised future never approaches 260k. The error is global because the claim is made about the overall regime, not a local segment.

### F.2. Task B: Global Price of a Standard Industrial Raw Material

**Background.** This series tracks the quarterly global market price of a Standard Industrial Raw Material, sampled from 1990 Q1 to 2025 Q2.

**Ground-truth Supporting Evidence.** The task carries two human-labeled causal chains, one explaining the history (HC1) and one specifying the projected future (HC2), both are anchored to a shared background event (E1). Distractor analyses below should be read against this ground truth.

---

**Task B: Standard Industrial Raw Material price**

```
History mechanism

1. In the early 1990s, the market for a Standard Industrial Raw Material faced a severe primary production
deficit as strategic inventories were depleted faster than new extraction operations could be commissioned.
Primary global production remained consistently below annual industrial requirements, forcing the sector to
rely on finite discretionary inventories that eventually reached critical lows.
2. The exhaustion of these stockpiles triggered a phase of panic-buying by industrial utilities, creating a
speculative environment where buyers competed for limited uncommitted supply.
3. This led to a realized speculative price bubble peaking at approximately $120/lb in late 1995.
4. The peak was broken when a large-scale conversion program, launched in 1993, began successfully
repurposing decommissioned materials into commercial fuel from around 1995, flooding the market with
secondary supply.
5. Followed by a rapid market correction back toward $30/lb by March, 1998.

Forecast mechanism

1. The Standard Industrial Raw Material sector in late 1998 entered a phase of severe supply-side
rationalization, as market prices dropped below the marginal cost of production for global Tier-1 extraction
assets.
2. Persistent low prices are expected to force major operators to suspend high-cost production facilities,
significantly tightening the available primary supply in the global market.
3. A freeze in exploration capital expenditure during the market slump is projected to prevent the
development of a new project pipeline; these extraction sites typically require 7-10 years from discovery to
first production due to the complexity of regulatory licensing, geological confirmation drilling, and
facility construction, meaning that even a future price recovery cannot quickly unlock new supply to meet
growing industrial demand.
4. As secondary supplies from conversion programs are projected to stabilize, the market is expected to
recognize an impending structural deficit triggered by steady growth in reactor requirements.
5. That tightening forward balance, built by facility suspensions and the absence of new project development,
is expected to draw utilities back into aggressive long-term contracting, lifting prices sharply toward
approximately $80/lb by late 2001.
6. We anticipate a projected non-linear recovery, with prices finding a cyclical floor at $20/lb in 1999
before trending upward toward a secondary peak of approximately $80/lb by late 2001.
```

---

**Time-series,** $segment \times trend$**.**

> "Following a phase of sharp price corrections, the sector has transitioned into a definitive **rounding-base behavior** ... the sector is projected to bypass previously anticipated lower cyclical floors ... a direct and sustained climb toward previous price highs ... recovery will be **linear rather than erratic**."

The document misreads the directional behavior of the late-history segment: it claims the most recent prices have already turned upward in a smooth rounding-base shape, justifying a linear recovery that bypasses any further dip. Empirically, the last 20 historical observations slope downward, with terminal value 31.20 below the window mean, and the realised future first dips to a low of 19.76 before the broader recovery, contradicting the "linear, no further downside" claim. The error is local to a segment: it is a misstatement of the recent direction, not of the global regime.

**Time-series,** $segment \times value$**.**

> "... the Standard Industrial Raw Material sector has reached a **definitive structural floor** ... Recent price action has consistently validated the marginal cost level ... prices are expected to enter an immediate and sustained recovery phase ... **without further downside volatility**."

The document misstates the value level of the late-history segment by asserting that a structural floor has been confirmed at the marginal-cost level. The realised future violates this in two ways: prices fall further to 19.76, well below the value range the document treats as the confirmed support, and the subsequent recovery is volatile rather than monotone, with the future reaching 77.83 after first dipping. The misread is segment-level because it concerns the value of a local support region, not a claim about the global level of the entire history.

## G. Context Conditions

We evaluate forecasters under eight context conditions to separate the effect of background information, ground-truth supporting evidence, raw supporting documents, and DR-generated outputs. These conditions differ only in the textual input supplied to the forecaster, the historical time series and forecast horizon are held fixed.

- **No Context (None).** The forecaster receives only the historical time series and forecast request, with no textual context.
- **Background (Bg.).** The forecaster receives a short generic description of the time-series variable, such as the measured quantity and unit, but no task-specific future-relevant information.
- **Original Context (Orig.).** The forecaster receives the original context from the source CAF instance before our task-generation transformations.
- **Rewritten Context (Rewrite).** The forecaster receives the context after entity disambiguation and time-shifting in the task-generation pipeline. This condition tests whether these transformations preserve the forecast-relevant content of the original context.
- **Supporting Evidence (Evidence).** The forecaster receives the ground-truth supporting evidence used to evaluate the DR stage. This contains the specific forecast-relevant content without the surrounding documents.
- **Supporting Documents (Supp. docs).** The forecaster receives the concatenation of all supporting documents for the task. This condition tests whether a forecaster can extract and assemble supporting evidence directly from raw documents.
- **DR-synthesized Evidence (DR).** The forecaster receives the evidence synthesized by a DR agent from the corpus. This condition reflects the raw output of the DR stage before any additional verification or filtering.
- **Verified DR (Verif. DR).** The forecaster receives the subset of DR-synthesized evidence that passes verification against the supporting evidence and distractor labels. This condition isolates the effect of filtering distractors and unsupported claims from the DR output.

The following shows an example of these eight context conditions for the task we used for DR failure mode analysis in §K.

---

**Example context conditions**

```
## No Context (None)

```
```

---

## Background (Bg.)

```
This time series records the electricity usage (in Watt, W) of a household.
```

---

## Original Context (Orig.)

```
This time series records the electricity usage (in Watt, W) of a household. The readings were recorded from
2025-10-06 00:00:00 to 2025-10-19 19:00:00, and the recording frequency is hourly (H). Initially, the owner
started renovation from 2025-10-12 18:00:00 to 2025-10-13 00:00:00 and did not stay at the unit, leading to
low power usage of the first 169 readings. After the renovation finished on 2025-10-13 01:00:00, the house
is fully occupied, leading to a permanent increase in electricity usage.
Scenario: Forecast the future electricity usage of this household in the next 180 hours, from 2025-10-19
20:00:00 to 2025-10-27 07:00:00.
```

---
```

## Rewritten Context (Rewrite)

```
This time series records the electricity usage (in Watt, W) of Riverside Cottage 12. The readings were
recorded from 2025-10-06 00:00:00 to 2025-10-19 19:00:00, and the recording frequency is hourly (H).
Initially, the owner started renovation from 2025-10-12 18:00:00 to 2025-10-13 00:00:00 and did not stay at
the unit, leading to low power usage of the first 169 readings. After the renovation finished on 2025-10-13
01:00:00, this household is fully occupied, leading to a permanent increase in electricity usage.
Scenario: Forecast the future electricity usage of Riverside Cottage 12 in the next 180 hours, from
2025-10-19 20:00:00 to 2025-10-27 07:00:00.
```

---

## Supporting Evidence (Evidence)

```
1. This time series records the electricity usage (in Watt, W) of a household. The readings were recorded
from 2025-10-06 00:00:00 to 2025-10-19 19:00:00, and the recording frequency is hourly (H).
2. the owner started rennovation from 2025-10-12 18:00:00 to 2025-10-13 00:00:00 and did not stay at the
unit
3. leading to low power usage of the first 169 readings
4. After the renovation finished on 2025-10-13 01:00:00, this household is fully occupied
5. leading to a permanent increase in electricity usage
```

---

## Supporting Documents (Supp. docs)

The full mode concatenates all 14 supporting documents (about 25k characters total).

---

## DR-synthesized evidence (DR)

```
This time series records the electricity usage (in Watt, W) of the household Riverside Cottage 12.

- Riverside Cottage 12 electricity usage is documented as Watt-based household power consumption, with
specific focus on October 2025 load behavior and occupancy-driven baseline changes. (doc_7245, doc_7271)
- During the renovation and heavy-labor phase, the owner was not occupying the premises and major appliances
and climate-control systems were deactivated, resulting in Wattage reductions due to a lack of human
activity. (doc_7236, doc_7238)
- Construction concluded on October 13, at which point living quarters became available and the household
reached full occupancy; this elevated post-handover baseline persisted through at least October 19.
(doc_7241, doc_7242, doc_7243, doc_7246)
- Full occupancy is the primary driver of ongoing electricity use, linked to continuous environmental
support, active residential services, and the simultaneous operation of essential systems that were formerly
inactive. (doc_7244, doc_7245)
- Post-occupancy electricity use in Cottage 12 is rising, characterized by a drifting higher daily baseline
and usage scaling through steady upward momentum. (doc_7259, doc_7260, doc_7261, doc_7274, doc_7275)
- A smart billing meter was installed on October 9, coinciding with a change in utility provider, followed
by a safety inspection on October 15 after equipment updates. (doc_7250)
- A smart-meter calibration window from October 10 to October 15 caused temporary reporting-metric shifts
and recorded spikes attributed to automated diagnostics rather than occupancy. (doc_7262)
- A localized storm and regional grid fluctuations between October 8 and October 14 caused abnormal high
readings while surge protectors remained active. (doc_7263)
- Security and lighting installation from October 7 to October 12 introduced transient peak consumption due
to high-frequency motion sensors and testing-related loads. (doc_7264)
- A strict energy quota system was implemented months before October 2025 to regulate peak demand and
localized grid capacity, which may constrain extreme usage even under full occupancy. (doc_7258)
- Localized environmental monitoring from October 10 to October 14 recorded humidity variation, changing
cloud cover, low nighttime light, and the presence of a reflective roof system designed to reduce heat gain.
(doc_7251)
- The cottage features smart-home controls for temperature, lighting, security, media, and whole-home audio,
which allows occupied behavior to affect both standby and active domestic electrical loads. (doc_7267)
- Logistics records for furniture and staging are inconsistent, with one document scheduling a curated
furniture package for October 18 and another listing deliveries for late October. (doc_7252, doc_7265)
- There is conflicting evidence regarding usage trends: one report suggests usage is gradually declining
after a post-renovation surge, while others describe sustained upward momentum after full occupancy.
(doc_7276, doc_7259, doc_7260, doc_7275)
- Some documents describe a stable low-energy or flat profile with no October occupancy changes,
contradicting reports of a post-October-13 elevated baseline following handover. (doc_7271, doc_7272,
doc_7273, doc_7241, doc_7246)
```

```
      - Several documents with matching date ranges refer to other properties (Meadowview Cabin 4, Willow Creek
      Suite 22, Lakeview Cabin 4, Suite 8, and Meadow View Studio 2) rather than Riverside Cottage 12. (doc_7253,
      doc_7254, doc_7268, doc_7269, doc_7270)
      - Evidence supports modeling a structural break in electricity usage from renovation/vacancy conditions to a
      fully occupied residential baseline following the October 13 handover. (doc_7238, doc_7241, doc_7242,
      doc_7246)
      - Within the occupied regime, the directional signal indicates a higher and potentially rising baseline
      driven by residents' domestic routines and concurrent essential-system operation. (doc_7244, doc_7245,
      doc_7259, doc_7275)
      - Short-lived drivers such as meter diagnostics, storms, security testing, and logistics introduce
      uncertainty into the operational records. (doc_7262, doc_7263, doc_7264, doc_7252, doc_7265)
```

---

## Verified DR (Verif. DR)

```
(After verification, no Codex-synthesized evidence remains.)
```

# H. Implementation Details

**Common interface.** All five deep-research (DR) agents are evaluated through the same per-task interface. Each agent receives the historical time series, the forecast task instructions, and the task document corpus, which contains both supporting documents and distractor documents but does not expose their labels. Each agent produces two outputs: a markdown research report and a structured list of synthesized evidence items. Each synthesized evidence item contains a free-form claim and the identifiers of the source documents it cites. Downstream forecasters consume the structured evidence list under the `deepresearch_insights` context mode and the full report under the `deepresearch` context mode. The agents differ only in how they search, read, and synthesize information from the shared corpus.

**Deep-research agents.** All DR agents are prompted with the same DP forecasting prompt plus the following research instruction: "Search the local environment and gather the most relevant documents and evidence snippets that would help forecast the target time-series values." CODEX (OpenAI, 2026) wraps the OpenAI Codex command-line agent backed by GPT-5.5 at high reasoning effort. It runs an iterative tool-use loop in which the model autonomously reads documents, drafts partial summaries, revises its reasoning, and commits a final report with cited synthesized evidence. DRBENCH (Abaskohi et al., 2025) implements a search–summarize–synthesize cascade: it retrieves relevant passages, compresses retrieved documents into per-document briefs, and synthesizes evidence from those briefs. BENCH2FUTURE (Wildman et al., 2025) follows a ReAct-style trajectory, alternating between search actions and observations over the corpus while maintaining a running outline of the forecast-relevant context. OPEN-DEEP-RESEARCH (LangChain AI, 2025) decomposes research into planning, retrieval through ReAct-style tool calls, and report writing. RETRIEVAL is a minimal non-agentic baseline that performs a single embedding-based retrieval pass followed by a one-shot synthesis call to extract evidence items, without iterative tool use. Codex uses GPT-5.5 as its underlying model with High reasoning efforts, the other four DR agents use Gemini-3 Flash.

**Forecasters.** We evaluate four families of forecasters. Statistical baselines include ARIMA, ETS, SES, and Naive forecasting. Pretrained time-series models include Chronos. Multimodal and time-series–language models include Aurora and TimeOmni-7B. Zero-shot LLM forecasters use the Direct Prompt (DP) forecasting strategy (Williams et al., 2025) with Gemini-3.1-flash-lite, Mistral-medium-3.1, Qwen-3.5 at 4B/9B/27B (Qwen Team, 2026), Llama-3.2-3B (Grattafiori et al., 2024), and Phi-4-mini (Abdin et al., 2024). We also evaluate MoiraiAgent, a forecasting agent that combines a Moirai forecasting backbone with a Gemini language-model component.

**Context conditions.** For end-to-end evaluation, the forecaster receives the evidence synthesized by a DR agent. For controlled CAF analysis, we additionally evaluate forecasters under the context conditions defined in § G, including No Context, Background, Original Context, Rewritten Context, Supporting Evidence, Supporting Documents, DR-synthesized Evidence, and Verified DR. These conditions allow us to separate failures due to missing supporting evidence, distractor contamination, and forecaster inability to use textual context.

**Infrastructure.** Closed-source language models, including GPT-5.5 for Codex and Gemini-3 Flash, Gemini-3.1-flash-lite, and Mistral-medium-3.1 for DR agents and zero-shot LLM forecasters, are accessed through OpenRouter[6]. Open-weight language models, including the Qwen-3.5 family, Llama-3.2-3B, and Phi-4-mini, are served locally through vLLM[7]. Time-series and multimodal forecasters, including Chronos, Aurora, and TimeOmni-7B, are also served locally. MoiraiAgent runs with its forecasting backbone served locally and its language-model component routed through OpenRouter. Unless otherwise stated, each forecaster draws $S=25$ forecast trajectories per task.

# I. Evaluation Metrics

## I.1. Deep Research Metrics

**Notation.** For each task $t$, let $\mathcal{E}_t$ denote the set of ground-truth supporting-evidence items evaluated for that task. For each supporting-evidence item $e \in \mathcal{E}_t$, let $R_e$ denote the set of supporting documents required to substantiate it. Each DR agent produces an ordered list of synthesized evidence items. We evaluate the top $K_t = |\mathcal{E}_t| + 5$ synthesized evidence items for task $t$. Let $M_e \in \{0, 1\}$ indicate whether the judge finds a semantic match between ground-truth supporting-evidence item $e$ and at least one synthesized evidence item in the top $K_t$. Let $C_e$ denote the set of source documents cited by the matched synthesized evidence item or items. For distractor metrics, let $Q_t$ denote the set of all resolved documents cited by the DR report, $D_t$ the set of distractor documents available for task $t$, and $P_t$ the full document pool available for task $t$.

**Evidence Recall.** Evidence Recall measures whether a DR agent recovers the ground-truth supporting evidence and cites the documents needed to support it:

$$\text{EvidenceRecall} = \frac{1}{\sum_t |\mathcal{E}_t|} \sum_t \sum_{e \in \mathcal{E}_t} M_e \frac{|R_e \cap C_e|}{|R_e|}.$$

This metric gives credit only when the synthesized evidence semantically matches a ground-truth supporting-evidence item, with partial credit for citing the required supporting documents.

**Supporting Document Recall.** Supporting Document Recall measures whether the agent locates the source documents that contain the ground-truth supporting evidence, independent of whether the synthesized evidence text is semantically correct:

$$\text{SuppDocRecall} = \frac{1}{\sum_t |\mathcal{E}_t|} \sum_t \sum_{e \in \mathcal{E}_t} \frac{|R_e \cap Q_t|}{|R_e|}.$$

This separates document-level retrieval from evidence-level synthesis.

**Distractor Avoidance.** Distractor Avoidance measures the fraction of cited documents that are not distractors:

$$\text{DistractorAvoidance} = 1 - \frac{|Q_t \cap D_t|}{|Q_t|}$$

for each task with at least one resolved citation, and is averaged across tasks. Higher values indicate that the agent cites fewer distractor documents. This metric is intentionally not normalized by the base rate of distractors in the corpus, since our goal is to measure the quality of the agent's actually cited evidence sources.

**Citation Volume.** We also track the average number of resolved document citations per task,

$$\text{CitationVolume} = \frac{1}{|\mathcal{T}|} \sum_t |Q_t|,$$

to distinguish selective agents from agents that cite broadly and thereby increase both supporting-document recall and distractor exposure.

## I.2. Forecasting Metrics

For each task, let $T$ be the forecast horizon, $S$ be the number of forecast samples, $\hat{y}_{s,t}$ the $s$-th forecast sample at horizon step $t$, and $y_t$ the ground-truth value. We normalize all errors by the mean absolute value of the target series over the forecast

---

[6] https://openrouter.ai
[7] https://github.com/vllm-project/vllm

horizon:

$$a \; = \; \left( \frac{1}{T} \sum_{t=1}^{T} |y_t| \right)^{-1}.$$

**Point forecast metrics.** The point forecast is the sample mean,

$$\bar{y}_t \; = \; \frac{1}{S} \sum_{s=1}^{S} \hat{y}_{s,t}.$$

We report scaled mean absolute error and scaled root mean squared error:

$$\text{sMAE} \; = \; a \cdot \frac{1}{T} \sum_{t=1}^{T} |\bar{y}_t - y_t|, \qquad \text{sRMSE} \; = \; a \cdot \sqrt{\frac{1}{T} \sum_{t=1}^{T} (\bar{y}_t - y_t)^2}.$$

**Distributional forecast metric.** For probabilistic accuracy, we compute the empirical Continuous Ranked Probability Score (CRPS) at each horizon step from the forecast samples:

$$\text{CRPS}_t \; = \; \frac{1}{S} \sum_{s=1}^{S} |\hat{y}_{s,t} - y_t| \; - \; \frac{1}{2S^2} \sum_{s=1}^{S} \sum_{s'=1}^{S} |\hat{y}_{s,t} - \hat{y}_{s',t}|.$$

We report the scaled CRPS averaged across the forecast horizon:

$$\text{sCRPS} \; = \; a \cdot \frac{1}{T} \sum_{t=1}^{T} \text{CRPS}_t.$$

Lower sCRPS indicates better distributional forecasts.

**Calibration and sharpness.** To assess probabilistic calibration, we report empirical 90% interval coverage:

$$\text{Cov}_{90} \; = \; \frac{1}{T} \sum_{t=1}^{T} \mathbf{1}[\hat{q}_{0.05,t} \leq y_t \leq \hat{q}_{0.95,t}],$$

where $\hat{q}_{p,t}$ denotes the empirical $p$-quantile across the $S$ samples at time step $t$. A well-calibrated forecaster should have $\text{Cov}_{90} \approx 0.9$. We pair coverage with scaled 90% interval width:

$$\text{sIW}_{90} \; = \; a \cdot \frac{1}{T} \sum_{t=1}^{T} \left( \hat{q}_{0.95,t} - \hat{q}_{0.05,t} \right).$$

Coverage measures calibration, while interval width measures sharpness.

**Aggregation across tasks.** For each forecaster–context condition, we report the mean metric value over tasks together with the standard error of the mean across tasks. Following standard winsorization for heavy-tailed benchmark distributions, per-task values of sMAE, sRMSE, and sCRPS exceeding 5 are clipped to 5 independently for each metric. The per-cell count of clipped tasks is reported alongside the metric. Coverage and interval width are bounded by construction and are not clipped. We also report an AVERAGE RANK metrics computed per task across all evaluated cells: ranking by sRMSE gives the point-estimate rank, and ranking by sCRPS gives the distributional rank.

## J. DR Agents Distractor Patterns

### J.1. In-Task vs Cross-Task Distractor Citation

The left panel of Figure 16 shows that distractor failures arise not only from misleading documents within the target task, but also from off-task documents retrieved from the shared corpus. DRBENCH, OPENDR, and the retrieval baseline cite a large number of cross-task distractors, indicating that they rely on semantic similarity without sufficiently grounding documents to the correct entity, horizon, and forecasting target. CODEX is more strongly in-task, which helps explain its higher evidence recall in Table 2, but it still cites many in-task distractors, showing that task grounding alone is insufficient. Agents must also reject documents that are temporally misaligned, causally irrelevant, or inconsistent with the observed series. This validates Dr-CiK's shared-corpus design: it separates coarse task-grounding failures from fine-grained distractor-rejection failures, making retrieval errors more diagnostic than in isolated per-task retrieval settings.

## J.2. Distractor Citation by Taxonomy

The right panel of Figure 16 shows a clear hardness hierarchy across distractor types. *Time-series* distractors are the dominant failure case for every agent, despite all subtypes having the same corpus size: they account for the largest share of citations across the board, and CODEX cites $81\%$ of the unique time-series distractor universe. In contrast, *profile* and *noisy* distractors are consistently avoided, with substantially lower recall for every agent, showing that they function as relatively easy negatives. This separation validates the design of our distractor taxonomy. Simple topical or entity-level distractors can be filtered by current agents, but forecast-dependent time-series distractors remain hard because they are both semantically plausible and numerically misleading: rejecting them requires checking whether the document's interpretation of the observed trajectory is consistent with the actual series and forecast horizon. This is precisely the capability that standard DR benchmarks do not test, and it explains why Dr-CiK exposes failures that would be hidden by generic hard-negative retrieval evaluation.

# K. DR Agent Failure-Mode Analysis

**Setup.** This task asks the agent to forecast the electricity usage of Riverside Cottage 12 over a 180-hour horizon (2025-10-19 20:00 to 2025-10-27 07:00). The historical window covers two weeks (2025-10-06 to 2025-10-19), during which the household underwent a brief renovation between the evening of October 12 and the early morning of October 13. The renovation forced the resident off-site, all major appliances were deactivated, and once construction concluded, the household became fully occupied with a permanently elevated baseline. The agent is given 14 supporting documents and 30 distractor documents, plus the time series itself. Codex reports 19 synthesized evidence items and cites 31 documents. None of the four non-trivial supporting evidence items (E2 through E5) is matched, so the supporting evidence recall on this task is zero.

**Supporting evidence.** The benchmark provides five items of supporting evidence for this task. E1 fixes the dataset metadata, E2 through E5 form the causal chain the agent must reconstruct.

> **Supporting evidence (Task 216)**
>
> ```
> E1. This time series records the electricity usage (in Watt, W) of Riverside Cottage 12. The readings were
> recorded from 2025-10-06 00:00:00 to 2025-10-19 19:00:00, and the recording frequency is hourly (H).
>
> E2. the owner started rennovation from 2025-10-12 18:00:00 to 2025-10-13 00:00:00 and did not stay at the
> unit
>
> E3. leading to low power usage of the first 169 readings
>
> E4. After the renovation finished on 2025-10-13 01:00:00, this household is fully occupied
>
> E5. leading to a permanent increase in electricity usage
> ```

## K.1. Codex-Synthesized Evidence

For reference, the 19 synthesized evidence items Codex produced are listed below. Distractor identification (items 14, 15, 16) is correct, and the qualitative structural-break framing (item 17) is on point, so the failure is not a document-retrieval or distractor-avoidance failure.

> **Codex-synthesized evidence (Task 216)**
>
> ```
> 1. Riverside Cottage 12 electricity usage is documented as Watt-based household power consumption, with
> specific focus on October 2025 load behavior and occupancy-driven baseline changes.
>
> 2. During the renovation and heavy-labor phase, the owner was not occupying the premises and major
> appliances and climate-control systems were deactivated, resulting in Wattage reductions due to a lack of
> human activity.
>
> 3. Construction concluded on October 13, at which point living quarters became available and the household
> reached full occupancy; this elevated post-handover baseline persisted through at least October 19.
>
> 4. Full occupancy is the primary driver of ongoing electricity use, linked to continuous environmental
> support, active residential services, and the simultaneous operation of essential systems that were formerly
> inactive.
> ```

**5.** Post-occupancy electricity use in Cottage 12 is rising, characterized by a drifting higher daily baseline and usage scaling through steady upward momentum.

**6.** A smart billing meter was installed on October 9, coinciding with a change in utility provider, followed by a safety inspection on October 15 after equipment updates.

**7.** A smart-meter calibration window from October 10 to October 15 caused temporary reporting-metric shifts and recorded spikes attributed to automated diagnostics rather than occupancy.

**8.** A localized storm and regional grid fluctuations between October 8 and October 14 caused abnormal high readings while surge protectors remained active.

**9.** Security and lighting installation from October 7 to October 12 introduced transient peak consumption due to high-frequency motion sensors and testing-related loads.

**10.** A strict energy quota system was implemented months before October 2025 to regulate peak demand and localized grid capacity, which may constrain extreme usage even under full occupancy.

**11.** Localized environmental monitoring from October 10 to October 14 recorded humidity variation, changing cloud cover, low nighttime light, and the presence of a reflective roof system designed to reduce heat gain.

**12.** The cottage features smart-home controls for temperature, lighting, security, media, and whole-home audio, which allows occupied behavior to affect both standby and active domestic electrical loads.

**13.** Logistics records for furniture and staging are inconsistent, with one document scheduling a curated furniture package for October 18 and another listing deliveries for late October.

**14.** There is conflicting evidence regarding usage trends: one report suggests usage is gradually declining after a post-renovation surge, while others describe sustained upward momentum after full occupancy.

**15.** Some documents describe a stable low-energy or flat profile with no October occupancy changes, contradicting reports of a post-October-13 elevated baseline following handover.

**16.** Several documents with matching date ranges refer to other properties (Meadowview Cabin 4, Willow Creek Suite 22, Lakeview Cabin 4, Suite 8, and Meadow View Studio 2) rather than Riverside Cottage 12.

**17.** Evidence supports modeling a structural break in electricity usage from renovation/vacancy conditions to a fully occupied residential baseline following the October 13 handover.

**18.** Within the occupied regime, the directional signal indicates a higher and potentially rising baseline driven by residents' domestic routines and concurrent essential-system operation.

**19.** Short-lived drivers such as meter diagnostics, storms, security testing, and logistics introduce uncertainty into the operational records.

## K.2. Item-by-Item Failure Analysis

SUPPORTING EVIDENCE E2: RENOVATION WINDOW

Two supporting docs make the renovation window explicit. The project schedule memorandum states that "the construction timeframe is set between 18:00 on October 12, 2025, and midnight the following day. Following the conclusion of these works on October 13, the residence will remain fully occupied with the new modernized baseline persisting continuously through the 19th of October and into the following week." The property-management notice to the resident is even more specific, scheduling the work "between 18:00 on October 12, 2025, and midnight the following day . . . the resident is required to find alternative overnight accommodation for the duration of this window." The closest codex-synthesized evidence is item 2, which states that during the "renovation and heavy-labor phase" the owner was not occupying the premises and major appliances were deactivated. The qualitative content is correct, but the explicit calendar-and-clock interval (18:00 on October 12 to midnight) has been replaced by an aspectual phrase ("renovation and heavy-labor phase"). The judge's verdict was that the synthesis "lacks the specific dates and times." The string 18:00 appears zero times in the entire codex report, despite being printed verbatim in two supporting docs.

SUPPORTING EVIDENCE E3: 169 LOW-USAGE READINGS

The count is not a derived quantity, it is stated outright in two supporting docs. The energy consumption analysis reports that "the software analysis aggregated the initial 169 logged entries for the period ending October 13th. . . . the facility experienced a significant decline in its load profile." The hardware operations summary is even more emphatic, repeating the count three times within four sentences: ". . . an isolated data set consisting of 169 logged entries. . . . the initial 169 logged entries associated with this hardware operations unit. . . . the resulting tally of these 169 specific readings reflected

a prolonged low-use state of the unit throughout the monitored period." Codex's nearest synthesis is again item 2, which abstracts the entire phenomenon into "Wattage reductions due to a lack of human activity." The specific count was discarded during synthesis. The judge concluded that "none of the predicted insights mention the specific quantity of '169 readings' or the specific low power usage associated with that exact count." The token `169` appears zero times in the codex report.

SUPPORTING EVIDENCE E4: PRECISE HANDOVER TIMESTAMP

The on-site project completion report supplies this detail with the same phrasing the codex evidence later borrows: "Structural upgrades and interior remodeling were officially concluded at 1:00 AM on Monday, October 13, 2025. All construction activities have ceased, and all protective barriers have been removed from the premises. ... The household is now fully occupied, and this residential status, along with the associated elevated power baseline, will persist continuously and indefinitely after October 13, 2025." Codex-synthesized evidence item 3 reads "Construction concluded on October 13, at which point living quarters became available and the household reached full occupancy, this elevated post-handover baseline persisted through at least October 19." The lexical alignment with the source is close enough to confirm that the agent did read this document and used it to construct this evidence item. Yet the `1:00 AM` timestamp present in the source has been collapsed to a calendar-day grain ("October 13"). The judge's verdict was that the synthesis "fails to include the specific timestamp (01:00:00)." Neither `01:00` nor `1:00 AM` appears in the codex report.

SUPPORTING EVIDENCE E5: PERMANENT VERSUS RISING

The facility-engineering assessment labors the point with three near-synonyms in a single paragraph: "Engineering observations confirm that the change in the residential population is enduring in nature. Consequently, the building is experiencing a permanent increase in total electricity usage due to this population change. These factors, stable occupancy and higher demand, have resulted in a lasting change to the overall power profile." Codex describes the same regime in two synthesized evidence items. Item 5 characterizes post-occupancy usage as "rising, characterized by a drifting higher daily baseline and usage scaling through steady upward momentum," and item 18 says the "directional signal indicates a higher and potentially rising baseline." Unlike E2 through E4 this is not a calendar-precision issue, it is a more subtle modal shift. The supporting evidence asserts a steady-state outcome (the baseline has stepped up and is expected to remain there). The synthesized evidence asserts a dynamic process (the baseline is still moving, with no commitment to whether it stabilizes or continues to drift). The two descriptions imply different forecast trajectories: a step function followed by a flat run versus a positive-slope continuous function. The judge's verdict was that the predicted insights "do not characterize this change as 'permanent'." The token `permanent` does not appear in the codex report.

## K.3. Distractor Docs Were Correctly Handled

The supporting evidence is supported by 14 supporting docs and is challenged by 30 distractor docs. Several distractor docs offer alternative explanations for the structural break observed in the time series. The utility maintenance log attributes the shift to firmware: "this variation is directly caused by a modification in the data logging logic integrated into the new firmware. This change represents a technical adjustment to reporting protocols rather than a shift in actual usage habits." The estate maintenance log introduces a billing-meter swap and a safety inspection in the relevant window: "On 2025-10-09, a smart billing meter was officially installed at Riverside Cottage 12. ... The local power company conducted a routine safety inspection at Riverside Cottage 12 on 2025-10-15 to ensure all systems meet current regulatory standards following the recent equipment updates." Codex picks these up as items 6 through 13 of its synthesis and explicitly flags them as conflicting in items 14, 15, and 16, including the wrong-property distractors that mention Meadowview Cabin 4 and Willow Creek Suite 22. In other words, the agent's distractor-resistance behavior on this task is exactly what the benchmark wants to see. The failure on E2 through E5 is therefore not attributable to distractor confusion.

## K.4. What the Failure Reveals

The four unmatched supporting evidence items share a single failure pattern, which is best described as *specificity collapse during synthesis*. In each case the agent retrieved the right supporting docs, identified the correct claim, separated it from competing distractor framings, and even preserved phrase-level lexical similarity in the synthesized evidence, what it discarded was the quantitative anchor printed in the supporting doc. The renovation interval 18:00 to midnight became "renovation and heavy-labor phase." The 169-reading low-usage segment, repeated three times in the hardware operations summary, became "Wattage reductions." The 1:00 AM handover moment, used verbatim in the completion report, became

"October 13." The adjective *permanent*, used three times in the facility-engineering assessment, became *rising*, *drifting higher*, and *steady upward momentum*.

The pattern is consistent with a generation-time preference for fluent analyst-style narrative over verbatim transcription of source specifics. Quoting the precise hour, the precise count, and the precise modal status of a baseline change reads as datasheet output rather than analytical synthesis, and the agent appears to default toward the latter style. For benchmarks and downstream tasks that consume the report as input to a forecasting model, this default is costly: the discarded specifics are precisely the anchors a forecaster needs to place a regime change at the right index, partition the training window correctly, and decide between a step-function extrapolation and a trend extrapolation. The failure is decoupled from retrieval quality, distractor resistance, and causal-chain construction, on each of these the agent was demonstrably correct on this task. It is a failure of the read-then-synthesize step, in which numeric anchors and modal qualifiers in the supporting docs are systematically replaced by aspectual or directional paraphrases in the synthesized evidence.

## L. Additional Context-Aided Forecasting Results

*Table 6.* Context-aided forecasting results on the 225-task subset of Dr-CiK where every listed model produced valid metrics. Values are mean ± sample standard deviation. Rows are grouped by context family: NO CONTEXT uses only the time series, DEEPRE-SEARCH EVIDENCE uses Codex-synthesized evidence, and ORIGINAL CONTEXT uses the source CAF context. Scaled error metrics are winsorized at 5.0 per task before aggregation. Models marked with $^*$ are direct-prompt LLM forecasters. `Gemini` denotes `gemini-3.1-flash-lite`, all *MoiraiAgent* variants use `Gemini`, and "+ Med." indicates medium reasoning effort.

| MODEL | AVG. RANK POINT | AVG. RANK DISTRIBUTION | SCALED MAE | SCALED RMSE | SCALED CRPS |
|---|---|---|---|---|---|
| NO CONTEXT | | | | | |
| ARIMA | $13.56 \pm 4.58$ | $13.09 \pm 4.78$ | $0.718 \pm 1.162$ | $0.867 \pm 1.174$ | $0.508 \pm 0.882$ |
| ETS | $13.46 \pm 5.93$ | $13.20 \pm 6.01$ | $0.524 \pm 0.532$ | $0.704 \pm 0.656$ | $0.431 \pm 0.431$ |
| Naive | $15.02 \pm 5.21$ | $15.28 \pm 5.09$ | $0.793 \pm 1.045$ | $0.941 \pm 1.075$ | $0.515 \pm 0.679$ |
| SES | $12.75 \pm 5.46$ | $12.50 \pm 5.56$ | $0.520 \pm 0.629$ | $0.700 \pm 0.723$ | $0.388 \pm 0.387$ |
| Chronos | $10.17 \pm 5.45$ | $9.01 \pm 5.40$ | $0.428 \pm 0.681$ | $0.632 \pm 0.839$ | $0.327 \pm 0.463$ |
| Aurora | $11.95 \pm 4.53$ | $12.46 \pm 4.82$ | $0.577 \pm 1.000$ | $0.763 \pm 1.092$ | $0.503 \pm 0.921$ |
| Gemini$^*$ | $10.40 \pm 4.99$ | $10.50 \pm 5.23$ | $0.377 \pm 0.564$ | $0.582 \pm 0.734$ | $0.327 \pm 0.545$ |
| Moirai Agent | $9.64 \pm 5.16$ | $11.48 \pm 5.38$ | $0.366 \pm 0.540$ | $0.560 \pm 0.722$ | $0.347 \pm 0.516$ |
| DEEPRESEARCH EVIDENCES (CODEX AGENT) | | | | | |
| Aurora | $12.07 \pm 4.47$ | $12.49 \pm 4.98$ | $0.579 \pm 1.002$ | $0.763 \pm 1.090$ | $0.503 \pm 0.918$ |
| Qwen3.5-4B$^*$ | $11.34 \pm 4.95$ | $9.88 \pm 4.81$ | $0.453 \pm 0.638$ | $0.640 \pm 0.834$ | $0.311 \pm 0.444$ |
| Qwen3.5-9B$^*$ | $11.24 \pm 5.17$ | $9.43 \pm 4.80$ | $0.432 \pm 0.604$ | $0.608 \pm 0.762$ | $0.296 \pm 0.412$ |
| Qwen3.5-27B$^*$ | $10.64 \pm 5.48$ | $9.73 \pm 5.46$ | $0.415 \pm 0.616$ | $0.572 \pm 0.778$ | $0.324 \pm 0.589$ |
| Mistral-Medium-3.1$^*$ | $11.62 \pm 5.68$ | $11.80 \pm 5.69$ | $0.447 \pm 0.640$ | $0.616 \pm 0.831$ | $0.380 \pm 0.622$ |
| Gemini$^*$ | $10.00 \pm 5.85$ | $10.28 \pm 5.97$ | $0.408 \pm 0.629$ | $0.567 \pm 0.796$ | $0.331 \pm 0.528$ |
| Gemini + Med.$^*$ | $9.83 \pm 5.55$ | $9.91 \pm 5.69$ | $0.377 \pm 0.554$ | $0.529 \pm 0.738$ | $0.321 \pm 0.532$ |
| Moirai Agent | $8.53 \pm 5.05$ | $9.21 \pm 5.00$ | $0.384 \pm 0.652$ | $0.528 \pm 0.798$ | $0.315 \pm 0.548$ |
| ORIGINAL CONTEXT | | | | | |
| Aurora | $12.19 \pm 4.47$ | $12.47 \pm 4.70$ | $0.580 \pm 1.007$ | $0.762 \pm 1.093$ | $0.507 \pm 0.932$ |
| Gemini$^*$ | $6.30 \pm 5.31$ | $6.54 \pm 5.47$ | $0.274 \pm 0.519$ | $0.393 \pm 0.680$ | $0.240 \pm 0.506$ |
| Moirai Agent | $4.58 \pm 3.72$ | $5.27 \pm 4.08$ | $0.248 \pm 0.498$ | $0.350 \pm 0.663$ | $0.212 \pm 0.472$ |
| Moirai Agent + Med. | $4.71 \pm 4.09$ | $5.46 \pm 4.21$ | $0.252 \pm 0.497$ | $0.355 \pm 0.664$ | $0.216 \pm 0.467$ |

Tables 7 and 6 report the same forecaster comparison under two complementary aggregation rules. Table 7 evaluates the full 240-task benchmark and reports mean ± standard error over the subset of tasks for which each model produced a valid metric. The FAIL column counts tasks excluded for that model because of runtime failures, non-finite outputs, or patched entries that were not recomputed. Table 6 instead restricts evaluation to the 225-task intersection on which every listed model produced a valid metric, enabling a head-to-head comparison without per-model task drift, it reports mean ± sample standard deviation. Both tables use the same winsorization rule: per-task sMAE, sRMSE, and sCRPS are

*Table 7.* Context-aided forecasting results on the full 240-task Dr-CiK benchmark. Values are mean $\pm$ standard error of the mean over the tasks for which each model produced valid metrics. The FAIL column counts tasks excluded for that model because of runtime failures, non-finite outputs, or patched entries that were not recomputed. Rows are grouped by context family: NO CONTEXT uses only the time series, DEEPRESEARCH EVIDENCE uses Codex-synthesized evidence, and ORIGINAL CONTEXT uses the source CAF context. Scaled error metrics are winsorized at 5.0 per task before aggregation. Bold marks the best result within each context family, plus results within $1.96\times$ standard error of the best. Models marked with $^*$ are direct-prompt LLM forecasters. `Gemini` denotes `gemini-3.1-flash-lite`, all *MoiraiAgent* variants use `Gemini`, and "+ Med." indicates medium reasoning effort.

| MODEL | AVG. RANK POINT | AVG. RANK DISTRIBUTION | SCALED MAE | SCALED RMSE | SCALED CRPS | FAIL |
|---|---|---|---|---|---|---|
| **NO CONTEXT** | | | | | | |
| ARIMA | $14.09 \pm 0.33$ | $13.43 \pm 0.34$ | $0.692 \pm 0.073$ | $0.834 \pm 0.074$ | $0.488 \pm 0.055$ | 0 |
| ETS | $13.88 \pm 0.42$ | $13.61 \pm 0.42$ | $0.507 \pm 0.034$ | $0.681 \pm 0.042$ | $0.418 \pm 0.027$ | 0 |
| Naive | $15.76 \pm 0.36$ | $15.91 \pm 0.36$ | $0.767 \pm 0.066$ | $0.910 \pm 0.068$ | $0.499 \pm 0.043$ | 0 |
| SES | $13.10 \pm 0.38$ | $12.77 \pm 0.38$ | $0.503 \pm 0.040$ | $0.677 \pm 0.046$ | $0.378 \pm 0.025$ | 0 |
| Chronos | $\mathbf{10.57 \pm 0.37}$ | $\mathbf{9.47 \pm 0.36}$ | $\mathbf{0.416 \pm 0.043}$ | $\mathbf{0.613 \pm 0.053}$ | $\mathbf{0.319 \pm 0.029}$ | 0 |
| Aurora | $12.28 \pm 0.33$ | $12.80 \pm 0.35$ | $0.554 \pm 0.063$ | $0.733 \pm 0.069$ | $0.483 \pm 0.058$ | 0 |
| Gemini$^*$ | $10.83 \pm 0.34$ | $10.99 \pm 0.35$ | $\mathbf{0.409 \pm 0.045}$ | $\mathbf{0.606 \pm 0.053}$ | $\mathbf{0.319 \pm 0.034}$ | 0 |
| MoiraiAgent | $\mathbf{9.86 \pm 0.35}$ | $11.80 \pm 0.37$ | $\mathbf{0.356 \pm 0.034}$ | $\mathbf{0.545 \pm 0.046}$ | $\mathbf{0.338 \pm 0.033}$ | 1 |
| **DEEPRESEARCH INSIGHTS** | | | | | | |
| Aurora | $12.41 \pm 0.32$ | $12.85 \pm 0.35$ | $0.556 \pm 0.063$ | $0.733 \pm 0.069$ | $0.483 \pm 0.058$ | 0 |
| TimeOmni-7B | $16.15 \pm 0.35$ | $18.15 \pm 0.30$ | $0.600 \pm 0.051$ | $0.801 \pm 0.058$ | $0.600 \pm 0.051$ | 23 |
| Llama3.2-3b$^*$ | $16.64 \pm 0.40$ | $15.28 \pm 0.42$ | $1.318 \pm 0.113$ | $1.496 \pm 0.113$ | $0.551 \pm 0.049$ | 36 |
| Phi4-mini$^*$ | $18.54 \pm 0.35$ | $16.43 \pm 0.43$ | $1.478 \pm 0.136$ | $1.633 \pm 0.141$ | $0.493 \pm 0.049$ | 31 |
| Qwen3.5-4b$^*$ | $11.88 \pm 0.35$ | $10.28 \pm 0.34$ | $0.457 \pm 0.042$ | $0.646 \pm 0.055$ | $\mathbf{0.314 \pm 0.029}$ | 12 |
| Qwen3.5-9b$^*$ | $11.72 \pm 0.37$ | $\mathbf{9.79 \pm 0.34}$ | $0.431 \pm 0.040$ | $0.610 \pm 0.050$ | $0.297 \pm 0.027$ | 11 |
| Qwen3.5-27b$^*$ | $11.09 \pm 0.39$ | $\mathbf{10.19 \pm 0.39}$ | $\mathbf{0.414 \pm 0.040}$ | $\mathbf{0.571 \pm 0.051}$ | $0.323 \pm 0.038$ | 11 |
| Mistral-medium-3.1$^*$ | $12.14 \pm 0.40$ | $12.27 \pm 0.40$ | $0.443 \pm 0.041$ | $0.607 \pm 0.054$ | $0.375 \pm 0.040$ | 8 |
| Gemini$^*$ | $10.61 \pm 0.41$ | $10.95 \pm 0.41$ | $\mathbf{0.455 \pm 0.049}$ | $\mathbf{0.620 \pm 0.059}$ | $\mathbf{0.326 \pm 0.033}$ | 0 |
| Gemini + med.$^*$ | $10.23 \pm 0.38$ | $10.40 \pm 0.39$ | $\mathbf{0.370 \pm 0.035}$ | $\mathbf{0.518 \pm 0.047}$ | $\mathbf{0.314 \pm 0.033}$ | 0 |
| MoiraiAgent | $\mathbf{8.90 \pm 0.35}$ | $\mathbf{9.62 \pm 0.35}$ | $\mathbf{0.375 \pm 0.041}$ | $\mathbf{0.516 \pm 0.051}$ | $\mathbf{0.310 \pm 0.035}$ | 1 |
| **ORIGINAL CONTEXT** | | | | | | |
| Aurora | $12.46 \pm 0.32$ | $12.81 \pm 0.34$ | $0.557 \pm 0.063$ | $0.732 \pm 0.069$ | $0.487 \pm 0.058$ | 0 |
| Gemini$^*$ | $6.55 \pm 0.36$ | $6.75 \pm 0.37$ | $\mathbf{0.289 \pm 0.038}$ | $\mathbf{0.411 \pm 0.048}$ | $\mathbf{0.233 \pm 0.032}$ | 0 |
| MoiraiAgent | $\mathbf{4.81 \pm 0.26}$ | $\mathbf{5.38 \pm 0.28}$ | $\mathbf{0.242 \pm 0.031}$ | $\mathbf{0.343 \pm 0.042}$ | $\mathbf{0.206 \pm 0.030}$ | 0 |
| MoiraiAgent + med. | $\mathbf{4.96 \pm 0.28}$ | $\mathbf{5.55 \pm 0.29}$ | $\mathbf{0.246 \pm 0.031}$ | $\mathbf{0.348 \pm 0.042}$ | $\mathbf{0.210 \pm 0.029}$ | 0 |

independently capped at $5.0$ before aggregation. The tables group forecasters into three context families: NO CONTEXT, where the forecaster receives only the time series, DEEPRESEARCH, where the forecaster receives Codex-synthesized evidence, and ORIGINAL CONTEXT, where the forecaster receives the source CAF context. Across both aggregation rules, the same qualitative pattern holds: high-quality supporting context improves DP-Gemini and MoiraiAgent substantially over no context, whereas Codex-synthesized evidence closes only part of this gap. Aurora remains largely context-insensitive, suggesting that its forecasts are dominated by the time-series modality rather than the textual context.

## M. Prompts

This appendix reproduces the prompts used by every LLM-driven step of Dr-CiK environment generation, the LLM judges that gate it, and the task-difficulty probes. Placeholders in curly braces (e.g. {context}, {task_id}) denote runtime fields that are filled in for each invocation.

## M.1. Generation Prompts

### M.1.1. ENTITY PIPELINE

---

**Entity Extraction (Stage 1)**

```
Here is a collection of background text from multiple forecasting tasks:

<background_corpus>
{background_corpus}
</background_corpus>

---

Can you help me extract the entities, the associated time-series variables, and the 'contain' relations
between them (only between extracted entities)?
For each task, there must be exactly one primary entity (the object directly associated with the time series
variable).

Please return JSON with this schema:
{
  "entities": [
    {
      "entity_name": "...",
      "time_series_variable": ["..."],
      "source_tasks": ["..."],
      "source_tasks_as_primary_entity": ["..."]
    }
  ],
  "contain_relations": [
    {
      "parent_entity": "...",
      "child_entity": "...",
      "evidence": "..."
    }
  ]
}

Requirements:
1. Entity list must be NON-REPEATED (deduplicated by real-world entity). The order in time_series_variable
should be the same as the order in source_tasks_as_primary_entity for the same entity.
2. The source_tasks_as_primary_entity should be the source_tasks that entities are directly associated with
the time series variable.
3. Entity names must refer to GENERAL real-world categories (type-level concepts). If an entity contains
descriptive modifiers (e.g., location, ownership, time, temporary context), REMOVE these modifiers and keep
only the base category name, and modifiers should not be extracted as entities.
4. Additional entities should only be extracted if they serve as parent entities that contain the primary
entity in a clear physical, geographical, or organizational sense. No additional entities should be
extracted otherwise.
5. Only include containment relations that are explicitly stated or universally true by definition. Do not
include contain_relations that require assumptions, background knowledge expansion, or multi-hop inference.
6. The parent_entity and the child_entity must be from the entity list. If no contain relation is clear,
return an empty list for "contain_relations".
7. Do NOT output transitive/ancestor containment edges. Only output the nearest parent (most specific
parent) for each child entity when multiple levels exist.
8. Only return a JSON object with the specified schema, and make sure it satisfies all the requirements
above. Do not include any additional text in the response.
```

---

**Entity Profile Generation (Stage 2)**

```
Now generate realistic but purely synthetic profiles for one target entity.

Target entity:
{entity_name}

Time-series variables for tasks where this target entity is the primary entity are:
{all_time_series_variables}

Please assign {k_profiles} different profiles to this target entity.

Please return JSON with this schema:
{
  "entity_name": "...",
  "profiles": [
    {
      "profile_id": "1",
      "name": "...",
```

```
      "details": {
        "...": "...",
      }
    }
  ]
}
```

Requirements:
**1.** The profiles must be realistic and detailed enough to sound like plausible real-world descriptions for the target entity.
**2.** The profiles must be purely synthetic. Do not copy or infer any factual background from the input tasks beyond the entity category itself.
**3.** The profiles must not contain any information that could affect forecasting for any of the listed time-series variables, either directly or indirectly.
**4.** Avoid predictive signals such as events, trends, operations, demand, weather, outages, anomalies, promotions, policies, schedules, maintenance, failures, holidays, traffic shifts, or any other factor that may influence the time series.
**5.** Keep the profiles as stable background descriptions only, the details should be realistic and just background information will be enough.
**6.** Generate exactly {k_profiles} profiles for the target entity.
**7.** The returned entity_name must match the target entity exactly.
**8.** Make sure the name and details are REALISTIC as if they were describing a real-world entity, but do not include any information that could be used to predict the time-series variables associated with the entity.
**9.** Only return a JSON object with the specified schema. Do not include any additional text in the response.

## Profile Assignment / Rewritten Background (Stage 3)

Original context: ```{original_context}```

Source task ID: ```{source_task_id}```

Primary entity: ```{primary_entity_name}```

Assigned primary entity profile: ```{primary_entity_profile}```

Ancestor entity chain from nearest parent to farthest ancestor: ```{ancestor_chain_profiles}```

---

You will rewrite one original context into a sample-level profiled version.

Rewrite instructions:

**1.** Replace the original entity mention with the assigned profile in a way that matches the original information density.
**2.** If the original context only mentions the entity as a name/category, only replace it with the generated profile name.
**3.** If the original context gives slightly richer but still stable identity/background information, you may use a small amount of profile detail, but do not increase the information density beyond the original context.
**4.** If ancestor entities are present in the original context, replace each mentioned ancestor using the assigned ancestor profiles as needed, again matching the original information density.
**5.** You can make the slight modifications needed to ensure the rewritten context is fluent and coherent, but do not change any of the original information beyond the entity profile replacement.

Please return JSON with this schema:
```
{
  "source_task_id": "...",
  "rewritten_context": "...",
  "background_rewritten_context": "...",
  "applied_profiles": {
    "primary_entity": {
      "entity_name": "...",
      "profile_id": "..."
    },
    "ancestor_entities": [
      {
        "entity_name": "...",
        "profile_id": "..."
      }
    ]
  }
}
```

Requirements:
**1.** The returned source_task must match the given source task ID exactly.
**2.** If there are no ancestor entities, return an empty list for applied_profiles.ancestor_entities.

**3.** Just replace the entity identity wording with the assigned profiles. Do not change any of remaining original context.
**4.** After rewriting, the reader should still be able to clearly identify the original primary entity type ({primary_entity_name}) from the rewritten context. If the entity type cannot be easily inferred from the context, you may explicitly mention the entity type at the relevant position(s) in the text.
**5.** Just replace the entity identity wording with the assigned profiles. Do not change any of remaining original context.
**6.** The `background_rewritten_context` should contain only the background information of the primary entity from the rewritten context, you shoudl mention the entity profile in it. No events, scenarios, or other information that is not background information of the primary entity.
**7.** The rewritten content must be fully synthetic and must not contain any real-world identifiable information (names, institutions, locations, events, etc.).
**8.** Only return a JSON object with the specified schema. Do not include any additional text in the response.

## M.1.2. GROUND-TRUTH EVIDENCE AND REASONING CHAIN

**Ground-Truth Evidence Extraction**

```` ```{context}``` ````

Extract the independent event(s) and the causal graph (each causal chain) from the text above, including any associated `time_start` and `time_end` (formatted as `YYYY-MM-DD HH:MM:SS`, if available). If a timestamp is not available, set its value to "null".

Requirements:
* Do not add any facts or context not explicitly stated in the text.
* When a causal relationship exists, represent the cause and the effect as two semantically complete events. Do not split discourse markers or partial clauses into separate events.
* Combine all background information, constraints, or other information with no causal relationship into a single event ("E1").
* Each event must contain the complete relevant original text span.
* Ensure that the extracted events collectively cover all the original information (no information is missing), and that no information is duplicated or overlapping.
* The provided content may include task instructions, often appearing in the last sentence. If present, ignore these instructions.
* If multiple statements describe successive states, value changes, or different aspects of the same underlying entity, process, or analogy, combine them into a single event (but not the background event) rather than splitting each state or timestamped change into separate events.
* An event should represent a complete semantic unit describing a state, action, or causal situation. Avoid splitting short phrases, discourse markers, or connective fragments (e.g., "Due to", "After", "resulting in") into separate events.
* The causal graph and causal context may be empty if the text contains no causal relationships. In this case, the objective is simply to decompose the original content into events and their corresponding text spans.
* Each element in the `causal_context` list must correspond one-to-one with an entry in the `causal_graph` list, and both lists must have the same length.
* The `causal_context` should only contain the original context covered by the events in the corresponding causal graph entry, do not add any other information, so different causal contexts should not be the same. But each causal context should be self-contained, expressing a complete meaning.
* If an event's `time_start` or `time_end` can be logically inferred from other events in the text, fill them in accordingly; otherwise set them to "null".
* If there is a gap between time intervals, do not merge time ranges across non-contiguous periods, represent them as separate events rather than combining them into a single continuous range. Make sure each event has a self-contained context.
* Do not treat system descriptions, design properties, or static characteristics (e.g., "is designed to", "scales linearly") as effects in a causal relationship unless they are explicitly caused by another event.
* Put only stable background or non-causal contextual information into E1. If the context states constraints on the time-series variable (e.g., bounds, maxima, minima, capacities, or other operating limits), extract those constraints as a separate self-contained event rather than merging them into E1.
* If an extracted event contains explicit timestamps or clearly timestamped transitions, do not leave its time fields as "null"; use the earliest and latest explicit timestamps covered by that event span, and split the event if merging would make the timing unclear.

Output format:
Return a valid JSON object with the following structure (and no additional text):

```
{
  "extracted_events": [
    {
      "event_id": "E1",
      "event": "complete relevant original text span, self-contained and semantically complete",
      "time_start": "YYYY-MM-DD HH:MM:SS or null",
      "time_end": "YYYY-MM-DD HH:MM:SS or null"
    },
    ...
  ],
  "causal_graph": [
```

```
    {
      "cause_event_id": "E1",
      "effect_event_id": "E2",
      "relation": "cause"
    },
    ...
  ],
  "causal_context": [
    "Natural language description of the causal relation using the original text, self contained, without
    omitting any information",
    ...
  ]
}
```

## Reasoning Chain Expansion

```
You are a reasoning-chain refactoring assistant.

Your task is to refactor the given input into a full reasoning chain by introducing exactly {hops} new
contexts.

Item type:
- {item_kind}
- source ids: {source_event_ids}

Goal:
Construct a coherent, step-by-step reasoning chain that preserves the original meaning while preventing
shortcut retrieval.
Also identify the materially real source-grounded details in the input and track where each one is preserved
in the expanded chain.

The final chain must:
- be logically valid and sequential,
- require step-by-step reasoning to traverse,
- avoid shortcut reasoning and retrieval,
- be locally obvious but globally non-obvious.

Core requirements:
1. Introduce exactly {hops} new contexts.
2. Preserve the original meaning of the input.
3. The output must be the full final chain (not only new contexts).
4. Original input contexts may be preserved, rewritten, split, or repositioned, but semantics must remain
unchanged; if the input uses descriptive wording to characterize a specific state or transition, preserve
that descriptive information for that same state or transition, and do not merge multiple descriptively
distinct changes into one broader statement.
5. Each context must be specific, standalone, and reasoning-relevant; do not use generic bridge steps, and
ensure every intermediate context preserves or advances the original causal specificity rather than
replacing it with a vague statement.
6. Adjacent contexts must have a clear causal (reasoning) connection.
7. The chain must be a single linear path (no cycles, no branching).

Shortcut prevention rules:
8. No single context or adjacent pair of contexts should be sufficient to reconstruct the final outcome.
9. Avoid placing critical or distinctive information at the beginning of the chain.
10. Distribute key information across NON-ADJACENT contexts.
11. Avoid repeating or clustering distinctive details in nearby contexts.
12. Ensure information can only be accessed through sequential traversal.

Identifier and anchor control:
13. Treat identifiers (timestamps, names, unique values, distinctive phrases) as controlled anchors.
14. Anchors must not be repeated, paraphrased, or redistributed in ways that enable shortcut matching.
15. Anchors must not allow any single edge (adjacent pair) to act as a shortcut.
16. Structured fields from input (e.g., time_start, time_end) are boundary constraints and must not be
freely propagated.

Detail distribution and rewriting:
17. Separate identifiers from descriptive details when possible.
18. Spread different details across NON-ADJACENT contexts.
    When direct separation would make the chain unnatural or incoherent, introduce intermediate context(s)
    to preserve realism while ensuring that such details remain SEPARATED by at least one reasoning step.
19. Rewrite original content with strong expression divergence:
    - All essential information must be preserved,
    - but the expression should be rewritten to be as different as possible in wording, phrasing, format
    (e.g., timestamps), and structure,
    - minimize lexical overlap and avoid distinctive or recognizable patterns,
    - the rewritten context should not be easily matched to the original through surface-level similarity,
    - prioritize maximal expression divergence while maintaining full semantic equivalence.
20. Do not produce contexts that closely mirror the original input.
```

```
Quality constraints:
21. Maintain realism and plausibility, and preserve temporal and logical consistency with the input,
including the original ordering, duration, persistence, reversals, and phase transitions of events or
states.
22. Preserve all essential information across the full chain.
23. Avoid trivial steps, redundancy, or artificial decomposition.

Interpretation rule:
- Use relation = "cause" for all edges (as general reasoning dependency).

Item-specific rules:
{item_specific_rules}

Output format:
Return either:
1. JSON with:
   - "source_detail_inventory"
   - "contexts"
   - "relation_graph"
   - "detail_coverage"
2. null

Alignment:
- `relation_graph` must contain exactly `len(contexts) - 1` edges.
- Each edge in `relation_graph` must include exactly one non-empty `relational_context`, so there must be
exactly `len(contexts) - 1` relational contexts in total.
- Each edge in `relation_graph` must also include `contains_source_grounded_details` and
`grounded_detail_ids`.
- Use `grounded_detail_ids` on an edge only when that edge preserves a real source-grounded relation/detail
rather than bridge-only reasoning.
- `detail_coverage[*].covered_by_edge_ids` must use the exact edge ids from `relation_graph`, formatted as
`R1->R2`, `R2->R3`, ... . Do not use any other separator.

Context rules:
- Use IDs R1, R2, ..., in order.
- `relation_graph` must use only these `context_id` values.
- Do not place original source ids directly inside `relation_graph`.
- The final chain must include both:
  - contexts newly introduced by the model, and
  - contexts derived from the original input.
- Each original source id must be represented by at least one context in the final chain.
- Each context must include:
  - context_id
  - context
  - origin: new | original_preserved | original_refactored
  - type_count: string in the form `<origin>_<n>`, where `n` is the 1-based occurrence index within that
  origin type as contexts appear in `contexts`
  - contains_source_grounded_details: true | false
  - grounded_detail_ids: list of `detail_id` values preserved in this context
- If origin is original_*, include source_context_id.
- If origin is new, do not include source_context_id.
- Count `new`, `original_preserved`, and `original_refactored` separately when assigning `type_count`. For
example, if the first two `new` contexts appear at `R2` and `R5`, their `type_count` values must be `new_1`
and `new_2`.
- Exactly {hops} contexts must have origin = "new".
- For identifiers originating from the input (e.g., timestamps, unique profiles, or distinctive references),
their associated semantic content and the identifier itself should be separated across different,
non-adjacent contexts whenever possible.
- The identifier and its corresponding descriptive information must not be co-located in a single context or
adjacent contexts in a way that enables direct retrieval.
- However, the chain must still allow the original information to be reconstructed through sequential
reasoning across multiple steps.
- Add `source_detail_inventory` before `contexts`.
- Add `detail_coverage` after `relation_graph`.
- Each context and each edge must indicate whether it contains any real source-grounded detail, and if so,
which `detail_id` values it preserves.
- Every materially real source-grounded detail from the input must appear in `source_detail_inventory` and
must be covered at least once in `detail_coverage`.
- Do not label inferred bridge reasoning as if it were a real source-grounded detail from the input.
- To maximize reasoning difficulty, distribute source-grounded details as sparsely as possible across
different, non-adjacent contexts or edges instead of clustering multiple details in one context whenever
fidelity and coherence allow.
- If a source-grounded detail is most naturally preserved in the transition between two contexts rather than
in a single context, it may appear only on the corresponding `relation_graph` edge; avoid duplicating the
same detail in a context unless necessary for faithfulness or coherence.

Fallback:
Return null if no meaningful, consistent reasoning chain can be constructed.
```

```
Output JSON schema:
{
  "source_detail_inventory": [
    {
      "detail_id": "D1",
      "detail": "source-grounded detail text"
    }
  ],
  "contexts": [
    {
      "context_id": "R1",
      "context": "...",
      "origin": "new | original_preserved | original_refactored",
      "type_count": "original_refactored_1",
      "source_context_id": "E1",
      "contains_source_grounded_details": true,
      "grounded_detail_ids": ["D1"]
    }
  ],
  "relation_graph": [
    {
      "source_context_id": "R1",
      "target_context_id": "R2",
      "relational_context": "...",
      "relation": "cause",
      "contains_source_grounded_details": false,
      "grounded_detail_ids": []
    }
  ],
  "detail_coverage": [
    {
      "detail_id": "D1",
      "covered_by_context_ids": ["R1"],
      "covered_by_edge_ids": []
    }
  ]
}

Input:
{context}
```

M.1.3. SUPPORTING DOCUMENT GENERATION

## Source Metadata Reconstruction

```
{context}

You are given one causal evidence bundle from a forecasting task.

Your job is to reconstruct exactly one plausible standalone real-world source that could directly contain
this information.

Requirements:

- Produce exactly one source metadata object.
- Use only information directly supported by the provided bundle.
- Do NOT add inferred background, broader causes, forecasts, or unstated implications.
- The source must be realistic, self-contained, and independently plausible.
- Prefer a source that naturally contains all of the information in this bundle with minimal extra scope.
- `provides` should contain self-contained atomic facts.
- `source_content_length` should be a realistic approximate target length for the final reconstructed source
content.
For calibration, typical ranges include: short alerts or log entries (~80-150 words), emails or internal
chat summaries (~40-120 words), internal memos or brief reports (~300-500 words), news articles or public
bulletins (~500-800 words), technical or operational reports (~550-600 words), policy documents or official
statements (~400-600 words), and longer formal reports, audits, or multi-section documents (~700+ words);
choose an appropriate range based on the source type and issuer rather than minimizing length.

Return valid JSON only with exactly this schema:

{
  "name": "...",
  "type": "...",
  "issuer": "...",
  "provides": ["..."],
  "notes": "Scope of this source and what it does NOT include",
  "source_content_length": "? words"
}
```

## Metadata-to-Document Content

```
{source_metadata}

You are a generator that reconstructs realistic source content from structured metadata.

Your task is to generate the original content of this source as it would appear in the real world.

---

Requirements:

- The content must match the real-world form of the source type (e.g., report, log, memo, webpage, dataset
description)
- Choose the most appropriate format automatically (e.g., plain text, structured log, table, markdown, or
HTML if applicable)
- Use realistic tone, structure, and formatting conventions
- All information in `provides` must be explicitly included
- Do NOT introduce new facts, assumptions, or inferred details
- If a "Reference background for consistency only" block is provided, use it only to keep entity/background
details consistent with the metadata, and do NOT write any content that is not already supported by the
structured metadata above
- If `source_content_length` is provided, use it as the target length guidance for the reconstructed source
content
- The length of the content should be appropriate and realistic for the given source type and any provided
`source_content_length`
- You MAY add minimal connective language for clarity and coherence
- The result must be self-contained and read like an authentic standalone artifact

---

Constraints:

- Do not mention the metadata or reconstruction process
- Do not add explanations outside the content

---

Output Format (JSON ONLY)

{
  "content": "full reconstructed source content"
```

```
    }
```

## M.1.4. DISTRACTOR DOCUMENT GENERATION

**Distractor Content Generation (Confounder / Noisy / Profile / Temporal)**

```
You are a distractor generation assistant.

Your task is to generate realistic but misleading or non-useful contexts (distractors) based on the given
input context.

Input:
```{context}```
{supplemental_context_block}

Goal:
Generate {num_distractors} distractor contexts of type `{distractor_type}` that:
- appear highly similar and plausible,
- share strong surface and structural similarity with the input,
- but do NOT support the correct reasoning path or final answer.

General requirements:
1. Distractors must be realistic and natural.
2. They must not be obviously wrong or nonsensical.
3. They must not allow recovery of the correct answer.
4. Distractors must not reuse or preserve any key contextual details from the input that are directly useful
for reasoning or prediction.
    - This includes events, time periods, details, causal relationships, causal_contexts, critical conditions,
    and outcome-related signals.
    - Distractors must not share the same underlying cause, mechanism, or predictive factors as the input.

5. Task relevance constraint (CRITICAL):
    - Distractors must be irrelevant or insufficient for solving the target task.
    - For forecasting tasks, distractors must not provide valid predictive signals about the future behavior
    or outcome of the target entity.
    - They may appear related, but should either:
      - lack key predictive information, or
      - contain misleading or non-applicable signals.

6. A human reader should be able to determine that the distractor is not useful or is insufficient.
7. The distractor must be fully self-contained:
    - it must read as a standalone document fragment,
    - it must not refer back to the original context, original event, or original time series,
    - and it must not use phrases like "the same", "as before", "the original event", or similar
    cross-references.

High-confusability requirement (IMPORTANT):
8. Distractors should preserve as much surface similarity as possible:
    - keep the same entities (e.g., names, locations, variables),
    - reuse similar wording and phrasing where appropriate (e.g., timestamps, structure, style),
    - maintain similar sentence patterns or structure.
    - Use plain text, no formatting, the content should be directly usable as a document body without any
    post-processing. Don't follow the format of the Input.

9. However, despite this similarity, the distractor must NOT become valid evidence.

Distractor type for this call:
- {distractor_type}: {distractor_type_description}

Type-specific requirements:
{distractor_type_requirements}

Diversity requirement:
- Avoid generating distractors that are too similar to each other.
- Vary the phrasing, emphasis, and document focus while keeping the same distractor type.

Critical constraint:
- No distractor should independently or jointly form a correct reasoning path.
- Every distractor in this call must use `distractor_type` = `{distractor_type}` exactly.

Forecasting-specific guidance:
- If the task involves prediction:
  - Distractors may include plausible but incorrect forecasts,
  - or correct patterns under different conditions,
  - but must not match the true scenario.

Output format:
```

```
Return a JSON object:

{
  "distractors": [
    {
      "provides": ["...", ...] # list of strings describing what to contain in the distractor content, no
      formatting, just plain text.
      "distractor_type": "{distractor_type}",
      "reason": "short explanation of why this is a distractor and how it differs from the original context"
    }
  ]
}
```

## Time-Series Distractor Specification

```
You are a distractor generation assistant.

Your task is to generate one realistic but misleading time-series distractor specification for a forecasting
task.

Input:

The history_values and future_values inputs are provided in direct-prompt time-series format, where each
line is `(timestamp, value)`.

history_values:
```{history_values}```

future_values:
```{future_values}```

context:
```{context}```

Goal:
Construct one distractor specification that:
- is realistic and plausible given the context,
- introduces a minimal mischaracterization of one time-series feature, expressed as a plausible but
misleading empirical pattern, tendency, or historical-style behavior, without using specific timestamps, and
remaining distinguishable from the true history when grounded in the data
- leads to a materially different future pattern,
- can only be identified as incorrect when grounded in the time series.

## Rules

1. Use the SAME context background. Do not change the entity and maybe the event type.
2. Do NOT introduce new causes or external mechanisms.
   - Only change how the history is characterized.
3. The distractor must NOT be easily rejected using text alone.
   - It should require time-series grounding to detect the error.
4. The distractor_history_description and distractor_future_pattern must be independent and self-contained.
The distractor should arise naturally from the mischaracterized pattern alone.

## Taxonomy

Valid scope-feature combinations:

- point: value
- segment: value, trend, periodicity
- global: value, trend, periodicity

## Allowed minimal errors by feature

For each selected (scope, feature_type), the distractor must apply ONE of the following minimal
modifications:

Value:
- point:
  - misstate the value at a specific time point (e.g., a wrong value, a small deviation, or a different
  local extremum)
- segment:
  - misstate the average level (e.g., describe as a wrong average value)
  - slightly misidentify local max or min (e.g., claim a peak is higher/lower than it is)
- global:
```

   – misstate the overall level (similar to segment but applied to the whole history)

Trend:
- segment:
  - misstate the local direction (e.g., describe as rising instead of flat, or flat instead of slightly decreasing)
- global:
  - misstate the overall trend (e.g., describe as increasing when it is roughly stable)

Periodicity:
- segment:
  - misstate whether periodicity is present (e.g., describe weak periodicity as absent, or wrong frequency)
- global:
  - misstate the overall periodic pattern (e.g., claim no periodicity when a weak pattern exists or wrong frequency)

## Procedure

Step 1: Inspect history_values, future_values, and context.

Step 2: Select one valid (scope, feature_type).

Step 3 : Reason about the true history pattern for that feature and how it leads to the true future pattern.
- Inside reasoning, also output suitable experience-style descriptions of the distracting pattern, such as empirical tendencies, observed patterns, or typical behaviors, rather than specific factual statements that could be directly checked against the time series.

Step 4: Write a concise ground-truth description of the history for that feature.

Step 5: Write a concise ground-truth description of the future pattern.

Step 6: Create a distractor history description by making an identifiable minimal error on ONLY the selected (scope, feature_type).
- A self-contained, plausible but misleading characterization of a different pattern narrated as a experience or potential scenario (with no specific timestamps) that would lead to a different future pattern.
- The error must be minimal: misstate that feature slightly, without changing the overall structure of the history.
- The error must be identifiable from the history itself.
- Keep all non-selected properties unchanged.
- Do not invent new patterns or exaggerate weak ones.
- The distractor's modified history description should not be stated as a false factual replacement of the true history. Instead, it should be written as a plausible but misleading description of the observed series, such as an empirical pattern, a typical tendency, or an expected mode of behavior.
- Do not describe the distractor history using specific, checkable facts about the real historical time series, such as exact timestamps, or explicit local events. These would risk creating conflicting information (a lie). Instead, express the distractor history as a plausible but misleading empirical tendency, pattern-level regularity, or conditional behavior (e.g., how the series typically behaves when values stay within a certain range or under a certain observed pattern).

Step 7: Derive a distractor future pattern from that incorrect history description.
- The distractor future must be a plausible but incorrect continuation under the same context.
- It must materially contradict the true future time series.
- When the context implies a future effect or consequence, the distractor future should also contradict that implication.
- The difference must be qualitative, not merely a small numeric deviation.

## Output format (STRICT JSON)

```
{
  "reasoning": "...",
  "scope": "...",
  "feature_type": "...",
  "gt_history_description": "...",
  "gt_future_pattern": "...",
  "distractor_history_description": "...",
  "distractor_future_pattern": "..."
}
```

### Time-Series Distractor Item

You are a distractor generation assistant.

Your task is to convert a time-series-based distractor specification into a realistic but misleading textual distractor context.

```
Input:

context:
```{context}```

time_series_mode:
```time_series_{scope}_{feature_type}```

distractor_history_description:
```{distractor_history_description}```

distractor_future_pattern:
```{distractor_future_pattern}```

Goal:
Generate 1 distractor context that:
- follows the incorrect time-series interpretation,
- is realistic and natural,
- preserves strong surface similarity with the original context, but with no specific timestamps,
- and remains misleading for solving the true forecasting task.

General requirements:
1. Your content must base only on distractor_history_description and distractor_future_pattern. The
distractor must be realistic and natural, and should be expressed using ONE of the following narrative
styles:

- conditioned:
  express the distractor as a conditional tendency grounded in the observed pattern
  (e.g., "when recent values stay within a moderate range without strong upward movement, the series is
  unlikely to produce a sharp spike",
         "under a relatively stable local segment, short-term fluctuations typically do not escalate into
         sustained peaks")

- summary:
  express the distractor as a concise but slightly biased characterization of the observed history
  (e.g., "recent values remain within a relatively narrow range with only mild variation, without a clear
  buildup toward a surge",
         "the series shows stable short-term fluctuations without a dominant upward or cyclical pattern")

- implication:
  express the distractor as an inferred future consequence of the observed pattern
  (e.g., "this pattern suggests that the upcoming period will remain within the current operating range
  rather than exhibit a sharp spike",
         "the observed behavior points to continued moderate variation instead of a pronounced peak event")

- comparative:
  express the distractor as a relative judgment between plausible interpretations
  (e.g., "the recent behavior appears more consistent with stable fluctuation than with a buildup toward a
  sharp peak",
         "the pattern is closer to a flat or weakly varying regime than to a strongly periodic or
         surge-driven structure")

Do not introduce specific structural assumptions that are not directly supported by the observed time series
(e.g., exact peak timing, fixed number of peaks, or rigid cycle forms). Prefer weakening or reframing an
existing pattern over specifying a new one.
The future description must be a natural consequence of the described time-series pattern.
**There should be a clear inferential connection between the provided characterization and the future
behavior.**
The future behavior should be framed as depending exactly on the described pattern, so that it reads as the
future pattern would follow if that characterization of the history is correct (e.g., phrased in a way
similar to "when this specific pattern occurs, the future tends to exhibit ..." or "under this pattern, the
future would show ...").
The distractor's modified history description should not be stated as a false factual replacement of the
true history. Instead, it should be written as a plausible but misleading description of the observed series,
such as an empirical pattern, a typical tendency, or an expected mode of behavior.
Avoid abrupt or unsupported transitions from history description to future claims.

2. It must not be obviously wrong or nonsensical.
3. In `context`, you can only use the background information, no any other details.
4. It must not allow recovery of the correct answer (scenarios described in the `context`).
5. It must remain faithful to the incorrect time-series interpretation.
6. It must contradict the true future implicitly through the wrong interpretation, not by explicitly stating
it is wrong.
7. Do not restate the original future-driving claim from the context when that claim would make the
distractor valid evidence for the true forecast.

High-confusability requirement:
8. However, despite this similarity, the distractor must NOT become valid evidence.
```

```
Output format:
Return a JSON object:

{
  "distractors": [
    {
      "provides": ["...", ...],
      "distractor_type": "timeseries",
      "reason": "short explanation of why this is a distractor and how it differs from the correct
      interpretation",
      "time_series_scope": "{scope}",
      "time_series_feature_type": "{feature_type}"
    }
  ]
}

The "provides" field should be a list of plain-text elements describing each detail the distractor content
contains, without any formatting or meta instructions. The content should be directly usable as a document
body without any post-processing. Do not follow the format of the Input in the "provides" content.
The "provides" part should base solely on the distractor_history_description and distractor_future_pattern
and the entity information.
```

## M.2. LLM Judge Prompts

### Entity Judge

```
Task ID: ```{task_id}```

Original Task Context:
```{context}```

Primary Entity Extraction Slice For This Task:
```{entity_extraction}```

Note:
- Each item in `primary_entities` is already restricted to the current task only.
- `time_series_variable_for_this_task` should list the variable(s) aligned to this exact task.
- Parent entities and contain relations are intentionally omitted from this validation step and are reviewed
separately.

You are validating whether the primary-entity extraction for this single task is correct enough for
downstream use.

Validation criteria:
1. There should be exactly one primary entity for this task.
2. The primary entity should be the object directly associated with the task's time-series variable.
3. The extracted primary entity should be a general entity category, not an overly specific named instance
or modifier-heavy variant.
4. `time_series_variable_for_this_task` should be present and correct for this task.
5. The slice should not include duplicate or irrelevant primary-entity candidates, and it should not miss a
clearly necessary primary entity.

Return valid JSON only with this schema:
{
  "judge": "correct | needs_correction | uncertain",
  "confidence": "high | medium | low",
  "summary": "short overall judgment",
  "issue_types": ["snake_case_issue_type"],
  "findings": [
    {
      "severity": "major | minor",
      "issue_type": "snake_case_issue_type",
      "target_id": "entity name, relation, or coverage",
      "reason": "short reason",
      "evidence": "short text span or explanation from the task context"
    }
  ],
  "correction_hint": "short note for a human editor"
}

Rules:
- If the slice is good enough for downstream use, set `judge` to `correct`.
- Use `needs_correction` when there is a clear problem.
- Use `uncertain` when the context is ambiguous or the extraction may be acceptable but you are not
confident.
- Keep `summary`, `reason`, `evidence`, and `correction_hint` concise.
- If there are no issues, return an empty list for `findings` and `issue_types`.
```

```
– Do not rewrite the extraction. Only validate it.
```

## Ground-Truth Evidence Judge

```
Task ID: ```{task_id}```

Original Context:
```{context}```

Ground Truth Evidences (events) Extraction Result:
```{gt_evidence_extraction}```

You are validating whether the ground-truth evidence extraction result for this single task is correct
enough for downstream use.

Important principle:
– The goal is NOT perfection, but usability.
– Minor differences in phrasing, granularity, or grouping that do NOT affect meaning or causal
interpretation should NOT be treated as errors.

Validation criteria:
1. Coverage:
   – All meaningful information and details from the context should be represented.
   – Minor omissions of low-importance details are acceptable if they do not affect understanding.

2. Event quality:
   – Events should be semantically complete and self-contained.
   – Slight differences in how events are grouped or phrased are acceptable if meaning is preserved.
   – Do NOT flag alternative but reasonable segmentations as errors.

3. Non-duplication:
   – Events should not unnecessarily repeat the same information.
   – Partial overlap is acceptable if it improves clarity.

4. Causal edges:
   – Causal relationships must be directly supported by the context.
   – Implicit but clear causal relations are acceptable.
   – Do NOT penalize missing weak or ambiguous causal links.
   – causal edges can be empty if the text contains no explicit causal relationships.

5. Background event (E1):
   – Should mainly contain non-causal or contextual information.
   – Minor misplacement of information into or out of E1 is acceptable if it does not affect causal
   structure.

6. Time fields:
   – Should match the context when explicitly available or inferable.
   – Missing or null timestamps are acceptable if not clearly stated or inferable.

7. Causal context alignment:
   – The `causal_context` list must correspond one-to-one with the `causal_graph`, and both lists must have
   the same length.

8. Constraints:
   Constraints over the time series variable should be treated as a SEPARATE event and should be
   self-contained; constraints refer to limits, or requirements imposed on the variable over time.

9. Consistency:
   The extracted events, causal_graph, and causal_context should be consistent, semantically aligned with
   one another, and should together preserve the overall meaning of the original context.

Judgment rules:

– "correct":
  – All 9 validation criteria are satisfied. (otherwise should be "needs_correction" or "uncertain")
  – Minor issues (stylistic differences, alternative valid segmentations, or low-impact omissions) are
  allowed as long as they do NOT affect:
     – semantic correctness
     – causal validity
     – overall usability for downstream tasks
  – No major issues are present.

– "needs_correction":
  – At least one major issue exists in any of the 9 validation criteria, including but not limited to:
     – missing key information that affects understanding (coverage)
     – incorrect, unsupported, or misleading causal relationships
     – inconsistent or misaligned events, causal_graph, and causal_context
```

```
        – if constraints are present, they should be correctly extracted as a separate event; if this is not the
          case, it is a major issue.
        – violations that break semantic correctness or usability

 – "uncertain":
     – The case is borderline, where:
         – multiple interpretations are plausible, AND
         – it is unclear whether the differences impact downstream usability
     – Use this sparingly; prefer "correct" if the extraction is usable.

Additional integrated guidelines:
 – Do NOT penalize alternative but reasonable event segmentation or phrasing.
 – Do NOT over-flag weak or ambiguous missing causal links.
 – Be strict only when issues affect meaning, correctness, or causal structure.
 – The granularity of events may vary: parallel or enumerated content can be grouped into a single event as
long as meaning and causal interpretation are preserved.
 – It's fine to have only a single event if there is no clear causal structure or if the text is best
represented that way.
 – Prefer usability over perfection.

Severity guidelines:
 – major: Affects correctness, meaning, or causal structure.
 – minor: Stylistic, formatting, or alternative-but-valid interpretation.

Return valid JSON only with this schema, no other text:
{
  "judge": "correct | needs_correction | uncertain",
  "confidence": "high | medium | low",
  "summary": "short overall judgment",
  "issue_types": ["snake_case_issue_type"],
  "findings": [
    {
      "severity": "major | minor",
      "issue_type": "snake_case_issue_type",
      "target_id": "event id, edge, or coverage",
      "reason": "short reason",
      "evidence": "short text span or explanation from the task context"
    },
    ...
  ],
  "correction_hint": "note for a human editor"
}
```

## Reasoning Chain Judge

```
Original Context:
```{context}```

Expanded Output To Validate:
```{expanded_output}```

Item type:
 – {item_kind}
 – source ids: {source_event_ids}

Note:
 – The expanded_output is either:
     – a JSON object with:
         – source_detail_inventory
         – contexts
         – relation_graph
         – detail_coverage
     – where each `relation_graph` element should carry its own `relational_context`
     – or null

You are validating whether the reasoning-chain refactoring result for this single item is correct enough for
downstream use.

Important principle:
 – The goal is NOT perfection, but usability.
 – Minor wording differences, slight structural variation, or alternative but reasonable decompositions
should NOT be treated as errors if the chain remains faithful, easy to follow, and usable downstream.

Validation criteria:
1. Valid fallback behavior:
     – Returning null is acceptable only when the original context does not support meaningful reasoning-chain
     refactoring.
```

**2.** Faithfulness:
  **-** The expanded chain may use a different form and may add reasonable intermediate reasoning steps, as long as it stays logically coherent and does not contradict or materially change the original meaning.
  **-** Some numbers may be rewritten in an unusual way, they are acceptable as long as they are logically equivalent and do not introduce ambiguity.
  **-** Before judging, enumerate each materially descriptive state/transition in the original input and verify that each one has a corresponding preserved description in the expanded output. If any descriptive occurrence is missing, merged away, or preserved only for a different transition, the output is not correct.
  **-** Do not omit any descriptive detail that characterizes a specific state or transition; if the original uses descriptive wording (e.g. rapid, smooth, gradual, abrupt, sustained, stable, sharp, brief, persistent, or similar) to specify how a particular change or state unfolds, that same descriptive information must be preserved for that same part of the chain, not merely approximated elsewhere.
  **-** For materially descriptive details, the expanded output must preserve the same number of descriptively characterized states/transitions as the original input, even if the wording is paraphrased; do not merge multiple descriptively distinct changes into a single less-specific description.

**3.** Coverage:
  **-** The expanded chain should preserve all important information from the original input, including important descriptive details for the corresponding parts of the chain, even if paraphrased.
  **-** If `source_detail_inventory` and `detail_coverage` are present, they should track source-grounded details consistently and should not label bridge-only reasoning as source-grounded preservation.

**4.** Local reasoning quality:
  **-** The chain should be easy to follow, and each adjacent step should have a clear and necessary reasoning connection.

**5.** No shortcut reasoning:
  **-** The chain should not make the final outcome too directly recoverable from the early steps, and removing an intermediate step should materially weaken the reasoning path.

**6.** Context quality and rewriting:
  **-** The contexts should be meaningfully rewritten, specific, non-trivial, and non-redundant rather than lightly edited copies of the original.

**7.** Equivalent reformulation:
  **-** Equivalent reformulations, unit conversions, or format changes are acceptable if the meaning is exactly preserved and no downstream ambiguity is introduced.

**8.** Overall usability:
  **-** The full chain should remain temporally and logically consistent with the original input, realistic enough, and usable for downstream tasks.

Judgment rules:

**-** "correct":
  **-** The chain is faithful, easy to follow, preserves the original meaning, does not drop important information, and has no major issue affecting downstream usability.

**-** "needs_correction":
  **-** At least one major issue exists, such as contradicting the original input, materially changing its meaning, dropping important information, introducing an unreasonable shortcut, or returning null when meaningful refactoring is clearly possible.
  **-** Reasonable elaborations or intermediate inferences by themselves are not errors.

**-** "uncertain":
  **-** The case is borderline and it is genuinely unclear whether the differences materially affect downstream usability.

Additional integrated guidelines:
**-** Prefer usability over perfection.
**-** Do not over-penalize alternative but reasonable reasoning decompositions.
**-** Be strict only when issues affect faithfulness, meaning preservation, reasoning-chain quality, or downstream usefulness.

Severity guidelines:
**-** major: Affects correctness, reasoning validity, shortcut safety, or preservation of meaning.
**-** minor: Stylistic, formatting, or alternative-but-valid interpretation.

Return valid JSON only with this schema, no other text:
```
{
  "judge": "correct | needs_correction | uncertain",
  "confidence": "high | medium | low",
  "summary": "short overall judgment",
  "issue_types": ["snake_case_issue_type"],
  "findings": [
    {
      "severity": "major | minor",
      "issue_type": "snake_case_issue_type",
      "target_id": "context id, edge, source detail, or null_decision",
```

```
      "reason": "short reason",
      "evidence": "short text span or explanation from the original context or expanded output"
    },
    ...
  ],
  "correction_hint": "note for a human editor"
}
```

## Supporting Documents Judge

```
Task ID: ```{task_id}```

Real Context:
```{context}```

Supporting Documents To Validate:
```{supporting_documents}```

You are validating whether the supporting documents for this single task are good enough for downstream use.

Important principle:
- Judge the supporting documents collectively.
- The goal is NOT perfection, but usability.
- Minor phrasing or formatting differences are acceptable if the documents still cover the useful details
from the real context.

Validation criteria:
1. Coverage:
   - Collectively, the supporting documents should contain all useful details from the real context.
   - Minor wording differences are acceptable if meaning is preserved.
   - Small omissions are acceptable only when they do not affect understanding or downstream usefulness.

2. Source realism:
   - The supporting documents should use appropriate source types and realistic presentation styles.
   - Metadata, tone, structure, and document format should feel plausible for the claimed source.

3. Context alignment:
   - The documents should stay aligned with the real context rather than drifting to a different scenario.
   - Do not treat clearly irrelevant extra content as acceptable.
   - No conflicting information: the documents should not contain information that contradicts the real
   context. (wrong time period, wrong entity, wrong setting, etc.)
   - If one supporting document presents itself as a final summary, final assessment, or precise forecast
   sequence for the target period, it should not omit a key transition that is part of the real context's
   expected sequence; such omissions are major only when they make the document materially misleading rather
   than merely incomplete on its own.

4. Temporal consistency:
   - The supporting documents should be temporally consistent with the real context.
   - Dates, time windows, and relative time expressions should not point to a clearly different incident or
   period.
   - If a document describes a different date or time period while presenting it as the same immediate or
   current event, that is a major issue.

5. Entity consistency:
   - The supporting documents must stay consistent about the core entity or asset identity referenced by the
   real context.
   - If the real context refers to a specific terminal, unit, site, device, location, or named entity, the
   supporting documents should not switch to a different one unless the equivalence is explicitly clear.
   - Cross-document inconsistencies in terminal names, unit IDs, site names, or other core identifiers are
   major issues when they make the evidence set refer to different entities.

6. Collective usefulness:
   - The documents may divide information across multiple sources, but together they should still be easy to
   use.
   - Redundant overlap is acceptable if it improves clarity, but excessive redundancy that reduces usability
   is a problem.

Judgment rules:

- "correct":
  - All 4 validation criteria are satisfied.
  - Minor issues are allowed if they do NOT materially affect coverage, realism, or downstream usefulness.

- "needs_correction":
  - At least one major issue exists, including but not limited to:
    - missing useful real-context details that affect understanding,
    - inappropriate or unrealistic source types / presentation,
    - documents that drift away from the real context,
```

```
        – collectively unusable supporting coverage.
        – documents that point to a clearly different date, time window, or incident period than the real
        context,
        – conflicting terminal / unit / site / device / entity identifiers across supporting documents or
        against the real context,

– "uncertain":
    – The case is borderline and it is unclear whether the issues materially affect downstream usefulness.
    – Use this sparingly; prefer "correct" if the supporting set is usable.

Severity guidelines:
– major: Affects coverage, realism, alignment, temporal consistency, entity consistency, or downstream
usefulness.
– minor: Stylistic, formatting, or low-impact presentation issue.

Output formatting rules:
– When referring to a specific supporting document, always use the provided `id` field exactly as given,
such as `Supporting Document 04`.
– Do not invent shorthand forms like `Doc 04`, `04/05`, or other alternative labels.
– If a finding concerns multiple documents, list each exact document id explicitly, for example `Supporting
Document 04; Supporting Document 05`.
– Keep the same exact document ids in `target_id`, `reason`, and `evidence` whenever a finding is tied to
specific documents.
– For every finding, `repair_target_ids` should list the exact provided document ids that should be
regenerated to fix that issue.
– Even when an issue is collective, map it to the concrete supporting document ids that should be repaired
whenever possible instead of leaving it purely abstract.
– `repair_instruction` must be concrete, edit-ready, and specific enough that applying it would resolve the
full issue for the listed `repair_target_ids` without guesswork.
– Across all findings, the repair instructions should collectively address all major issues in the
supporting set.

Return valid JSON only with this schema, no other text:
{
  "judge": "correct | needs_correction | uncertain",
  "confidence": "high | medium | low",
  "summary": "short overall judgment",
  "findings": [
    {
      "severity": "major | minor",
      "issue_type": "snake_case_issue_type",
      "target_id": "exact provided document id such as Supporting Document 04, or a precise collective label
      such as collective_coverage or source_realism",
      "repair_target_ids": ["Supporting Document 04"],
      "reason": "short reason",
      "evidence": "short text span or explanation from the real context or supporting docs",
      "repair_instruction": "specific instruction that would fully resolve this issue for the listed
      repair_target_ids"
    }
  ],
  "correction_hint": "note for a human editor"
}
```

## Distractor Documents Judge

```
Task ID: ```{task_id}```

Real Context:
```{context}```

Distractor Type:
```{distractor_type}```

Distractor Documents To Validate:
```{distractor_documents}```

You are validating whether this distractor set for the given distractor type is good enough for downstream
use.

Important principle:
– Judge the distractor documents for this distractor type as a set.
– The goal is NOT perfection, but usability.
– Minor stylistic issues are acceptable if the distractor still works as a realistic, distinguishable, and
relevant distractor.

Validation criteria:
1. Realism:
    – The distractor should feel like a plausible real-world artifact.
```

– Source type, metadata, tone, and presentation should be realistic.

**2.** Distinguishability:
  – The distractor should remain distinguishable from the real context.
  – If it effectively restates or supports the real context, it is invalid.

**3.** Relevance:
  – The distractor should stay meaningfully related to the same task setting, entity, or scenario.
  – Completely unrelated distractors are not useful.

**4.** Type consistency:
  – The distractor should match the intended distractor type.
  – The content should reflect the stated distractor type rather than drifting into a different distractor pattern.

Type-specific rules:
{distractor_type_specific_rules}

Judgment rules:

– "correct":
  – All 4 validation criteria are satisfied.
  – Minor stylistic issues are allowed if the distractor remains realistic, distinguishable, relevant, and type-consistent.

– "needs_correction":
  – At least one major issue exists, including but not limited to:
    – unrealistic source or presentation,
    – distractor content that is not distinguishable from the real context,
    – distractor content that is not meaningfully relevant,
    – distractor content that does not match the intended distractor type.

– "uncertain":
  – The case is borderline and it is unclear whether the distractor is sufficiently distinguishable or type-consistent.
  – Use this sparingly; prefer "correct" if the distractor is usable.

Severity guidelines:
– major: Affects realism, distinguishability, relevance, or distractor-type consistency.
– minor: Stylistic, formatting, or low-impact presentation issue.

Output formatting rules:
– When referring to a specific distractor document, always use the provided `id` field exactly as given, such as `Distractor Document 01`.
– Do not invent shorthand forms like `Doc 01`, `01/02`, or other alternative labels.
– If a finding concerns multiple documents, list each exact document id explicitly, for example `Distractor Document 01; Distractor Document 02`.
– Keep the same exact document ids in `target_id`, `reason`, and `evidence` whenever a finding is tied to specific documents.

Return valid JSON only with this schema, no other text:
```
{
  "judge": "correct | needs_correction | uncertain",
  "confidence": "high | medium | low",
  "summary": "short overall judgment",
  "issue_types": ["snake_case_issue_type"],
  "findings": [
    {
      "severity": "major | minor",
      "issue_type": "snake_case_issue_type",
      "target_id": "exact provided document id such as Distractor Document 01, or a precise collective label
      such as distractor_type, realism, relevance, or distinguishability",
      "reason": "short reason",
      "evidence": "short text span or explanation from the real context or distractor docs"
    }
  ],
  "correction_hint": "note for a human editor"
}
```

## M.3. Task-Difficulty Labeling Prompts

### Certainty

You are an expert evaluator of causal reasoning chains. Your task is to classify each step of a causal chain by its certainty -- the epistemic status of the information it contains: how confident one can be that the information is accurate, precise, and reliable. Certainty arises from the method of measurement or estimation and the use of probabilistic language.

Levels:
  Certain: values are directly observed or measured with negligible error margins. No probabilistic language is used.
  Likely: at least one of the following applies: the step uses probabilistic language or confidence intervals (e.g. "likely", "70% probability"); values are modelled or estimated with acknowledged but bounded uncertainty.
  Uncertain: multiple sources of uncertainty compound -- for instance, a probabilistic projection with significant measurement error. Explicit uncertainty reasoning is required, or a judgment call about how much to trust the information must be made and justified.

Output only valid JSON, no explanation.

### Domain Knowledge

You are an expert evaluator of causal reasoning chains. Your task is to classify each step of a causal chain as General or Specialist, based on whether the causal edge -- the link between a trigger and its downstream effect -- can be traversed without domain-specific knowledge.

Definitions:
  General: the causal step can be traversed using common sense or broad world knowledge. No specialized expertise is required. Examples: heat causes increased air conditioning use; a public holiday reduces industrial activity.
  Specialist: the causal step requires domain-specific knowledge not available to a general audience, such as use of specialized lingo without definition. Examples: a specific industrial process shutdown reducing baseline grid load; a policy change in time-of-use pricing shifting demand timing; an infrastructure configuration affecting transmission capacity; talking of turbidity without saying it's an optical property.

Classify at the level of the individual causal edge -- each link between trigger and effect must be classified independently, regardless of how easy the surrounding steps are.

Output only valid JSON, no explanation.

### Explicitness

You are an expert evaluator of causal reasoning chains. Your task is to classify each step of a causal chain by its explicitness -- how clearly and centrally the relevant information is signaled within the step text. Explicitness combines three sub-components: the explicitness of the causal link, the salience of the trigger within the text, and the scope alignment between the step and the forecasting target (the quantity of interest).

Levels:
  Explicit: the step explicitly names the quantity of interest, directly states the trigger and its effect (including quantification where relevant), and the trigger is the main subject. No inference required.
  Implied: the step mentions the trigger but not its effect on the quantity of interest, or the connection is stated qualitatively without quantification. The trigger may be a secondary mention. Scope may require minor extrapolation (e.g. different location, same region).
  Implicit: the step describes a condition or event with no explicit connection to the quantity of interest. The link must be constructed through reasoning or inference. The trigger is peripheral or embedded in unrelated content. Scope requires meaningful extrapolation or transfer across geography, population, or context.

Output only valid JSON, no explanation.

### Temporal Complexity

You are an expert evaluator of causal reasoning chains. Your task is to classify each step of a causal chain by its temporal complexity -- the richness of temporal reasoning required to infer the quantitative effect of that causal step on the time series.

Temporal complexity has two independent sub-components:
  **1.** Delay and duration of the causal effect
  **2.** Periodicity: one-off event vs. recurring or compounding pattern

```
Levels:
  Straightforward: immediate effect, short and bounded duration, one-off event whose effect interval is easy
  to identify.
  Predictable: delayed effect (days to weeks), moderately variable duration, recurring but regular and
  predictable pattern (e.g. seasonal, weekly).
  Variable: long lag or cumulative, non-linear effect that unfolds over an extended trajectory, irregular or
  compounding recurrences that are hard to disentangle.

Output only valid JSON, no explanation.
```

## M.4. Final Agent Audit Prompt

**Final Agent Audit**

```
You are auditing the final exported benchmark supporting documents before forecasting.

Goal:
Find supporting documents that contain likely invalid, leaked, over-specific, or internally inconsistent
numeric claims, and prepare a human-reviewable correction plan. Do not modify files unless explicitly asked
later.

Scope:
- Inspect the exported benchmark tasks and documents under:
  env_generation/exports/final_v1/reasoning_hops_4/public
- Focus on supporting documents, not distractors.
- Check all tasks, not only a sample.
- Do not use model forecasting results or downstream performance to decide whether a document is valid.

What to Look For:
1. Leakage: documents that reveal exact future target values, exact forecast trajectories, deterministic
equations, or formulas that trivially determine the target.
2. Scale inconsistency: numeric claims that conflict with the actual observed history, future values, task
metadata, or unit scale.
3. Impossible or implausible values: wrong baselines, wrong units, impossible timestamps, inconsistent rates,
or trend magnitudes that are materially incompatible with the time series.
4. Over-specificity: semantically valid documents whose numeric details are too precise, wrong, or
unnecessary for the intended event.
5. Context risks: supporting documents that could mislead a forecaster or give a DR agent an invalid
shortcut.

Important distinction:
- Do not flag a document merely because it discusses a future event or qualitative future direction.
Supporting documents are allowed to contain forecast-relevant context.
- Flag it when it gives exact target values, target-window trajectories, deterministic formulas, or numeric
claims that are materially inconsistent with the realized time-series scale.
- Be pragmatic: do not over-police harmless qualitative claims. Flag only issues that could materially
affect forecasting or benchmark validity.

Audit Method:
For each task:
- Read the task context, `series.history_timestamps`, `series.history_values`, `series.future_timestamps`,
`series.future_values`, ROI if available, and all supporting documents.
- Do not rely only on the natural-language context; compare document claims against the actual time-series
values.
- If a document mentions specific values, rates, baselines, formulas, timestamps, or expected future
behavior, normalize units where needed and verify consistency with the observed history/future value scale.
- Check whether the document leaks exact future target values or makes the forecast mechanically
recoverable.
- Flag a major issue when a supporting document gives absolute target values, ranges, baselines, or formulas
that are materially outside the realized time-series scale or imply a different forecast trajectory, even if
the document is semantically aligned with the task context.
- Flag a minor issue when the document is semantically valid but includes unnecessary numeric precision that
should be softened to qualitative wording.

Severity:
Use one of:
- Major: likely affects benchmark validity, leaks target values, gives deterministic forecast information,
or conflicts materially with the time-series scale.
- Minor: wording is over-specific or slightly risky, but the intended event remains valid and the issue is
unlikely to change results substantially.
- Needs human judgment: ambiguous cases where the document may be valid, but the numeric claim should be
reviewed.

Output Requirements:
Produce a concise human-review report. For each flagged task, include:
```

- benchmark_id
- source task id if available
- severity
- problematic supporting document IDs
- short description of the issue
- why it is a real problem
- minimal recommended edit
- whether forecasting/deep-research results should be rerun after the edit

Correction Style:
Recommend minimal edits only. Prefer converting bad numeric claims into qualitative descriptions while preserving the intended event semantics.
Examples:
- Replace exact future values with "stable post-update baseline"
- Replace exact slope/formula with "steady linear deceleration"
- Replace incorrect baseline percentages with "normal operating baseline"
- Remove deterministic equations or target-window forecasts
- Replace overly precise timestamps with coarse event timing if exact timing is unnecessary

Do Not:
- Do not rewrite unrelated documents.
- Do not change time series values.
- Do not change task metadata.
- Do not change distractors unless they are mislabeled as supporting.
- Do not make large stylistic rewrites.
- Do not silently patch files.
- Do not optimize based on model performance.
- Do not use forecasting results to decide whether a document is valid.

Verification / Rerun Guidance:
After proposing fixes, identify which downstream artifacts would need rerun:
- forecasting runs for affected task/context modes
- deep research caches if supporting documents were already used
- review-server-visible cached contexts if applicable
- any exported benchmark bundle or metadata index containing the edited documents

Final Deliverable:
A structured list of flagged tasks and minimal fixes, ready for human approval before any file changes.
If no issues are found, report that explicitly and briefly describe the checks performed.

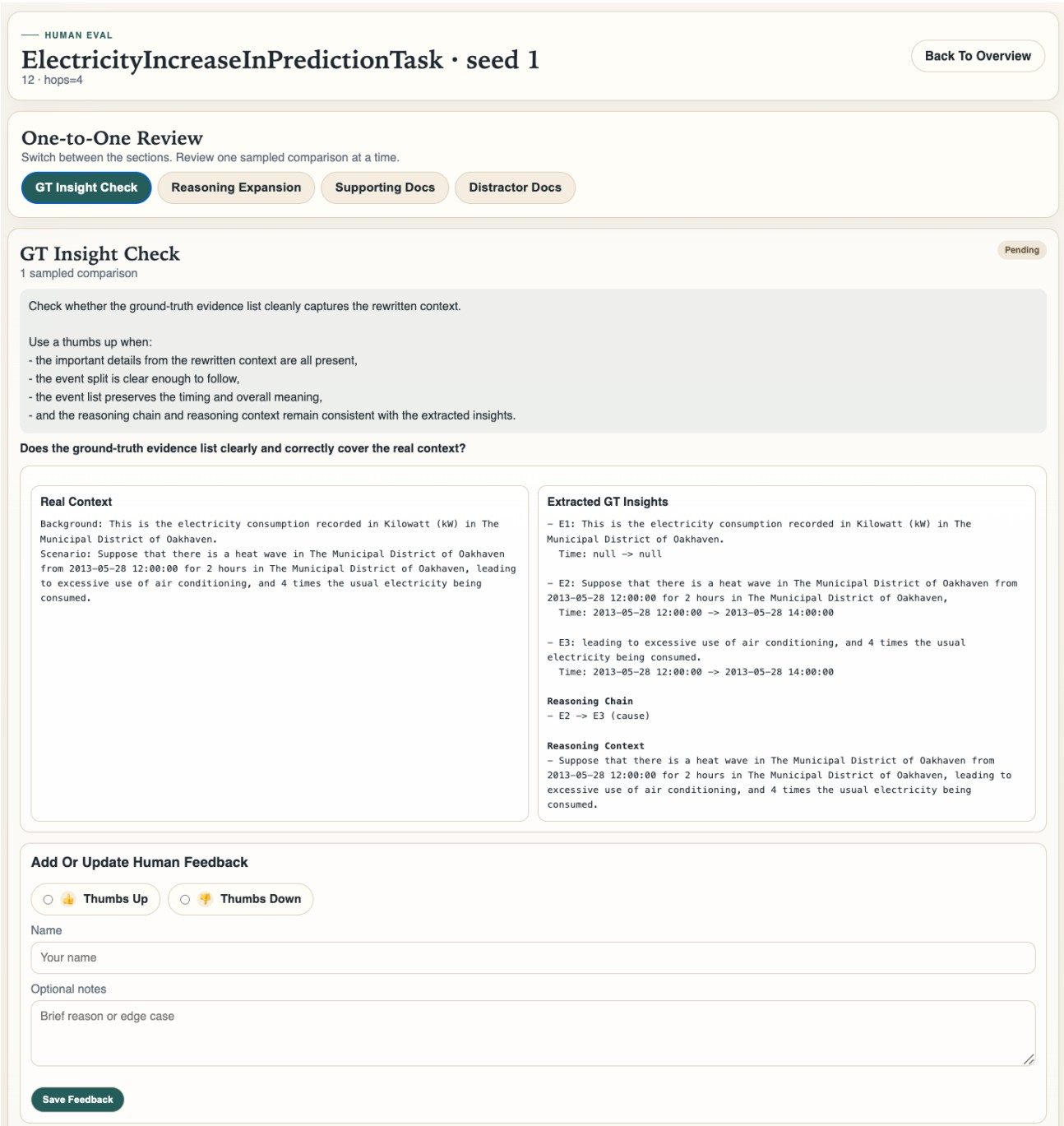

*Figure 10.* **Detailed human review interface.** For each sampled comparison, reviewers see the relevant source input and generated output side by side, along with the stage-specific review instructions. Reviewers mark the output as acceptable or unacceptable and may add notes explaining the failure mode. These annotations provide the human reference labels used to tune and select conservative LLM judges.

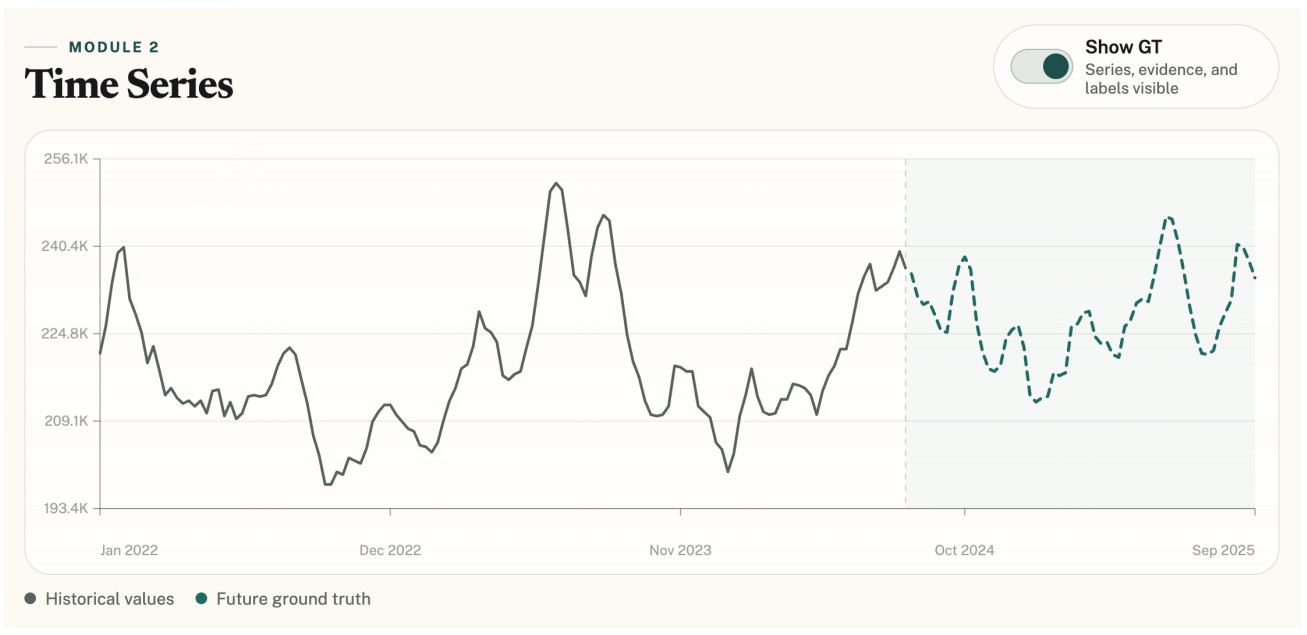

*Figure 11.* Task A: 4-week moving average of Initial Claims, weekly from January 2022 to September 2025.

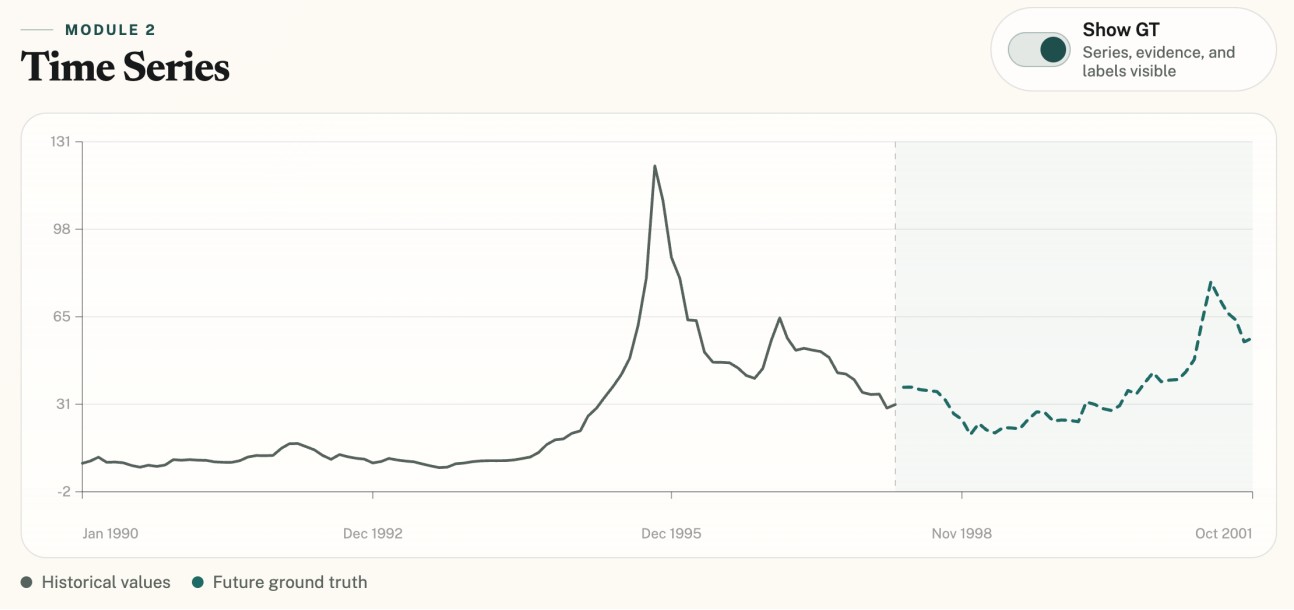

*Figure 12.* Task B: quarterly global price of a Standard Industrial Raw Material from 1990 Q1 to 2025 Q2.

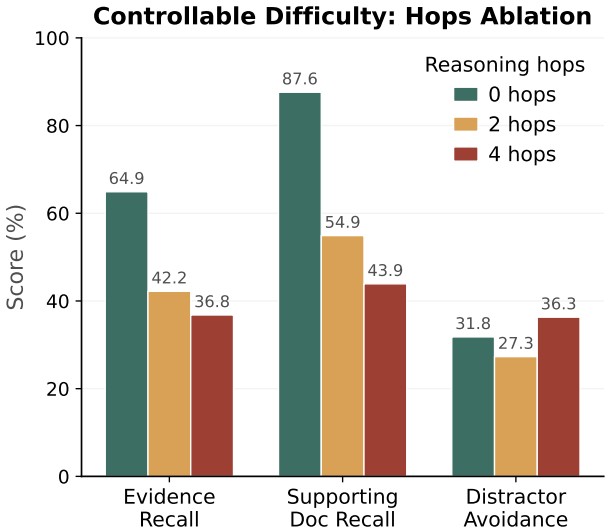

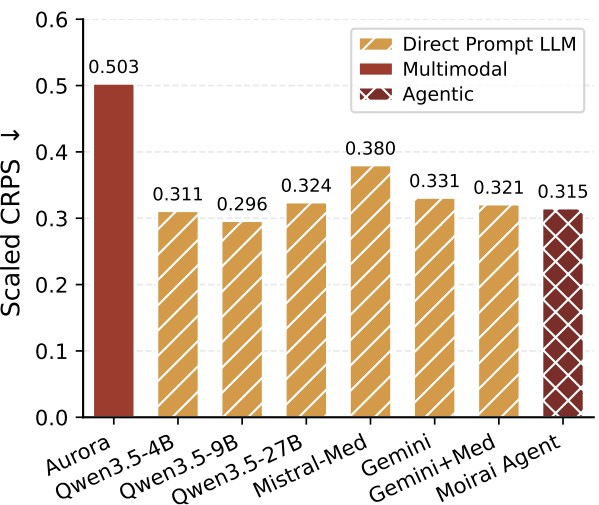

*Figure 13.* Evidence recall vs. number of reasoning hops. Recall degrades sharply as hop count grows, confirming that multi-hop synthesis is the binding constraint for DR agents.

*Figure 14.* Forecaster comparison under Codex-synthesized evidence. MoiraiAgent and DP-Qwen3.5 lead; scaling Qwen3.5 from 4B to 27B shows no improvement.

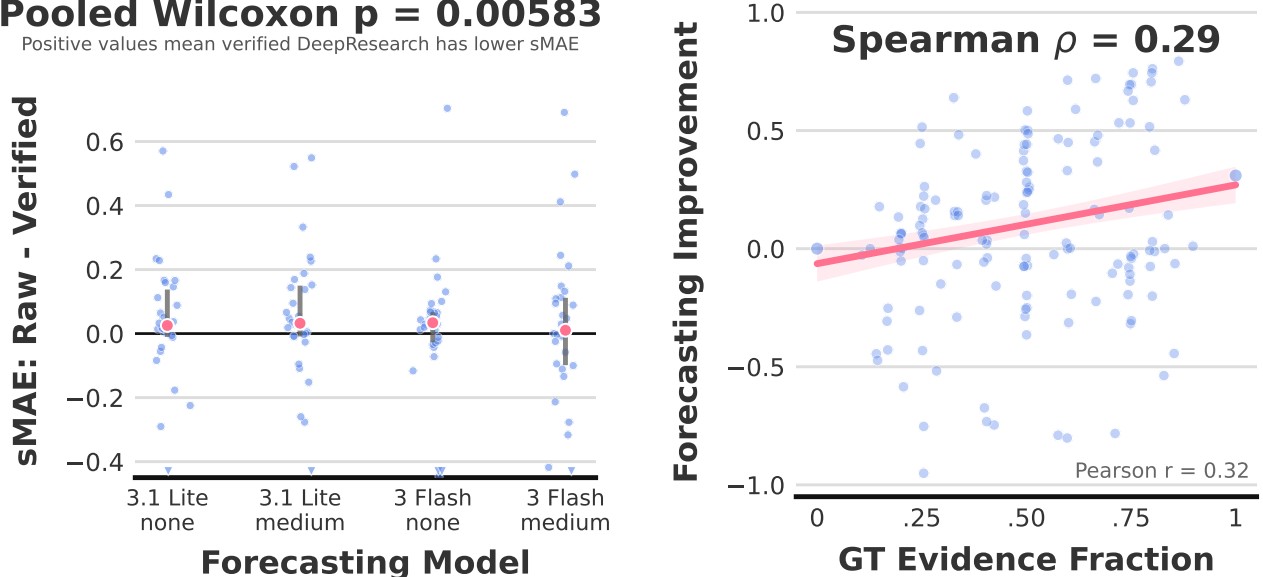

*Figure 15.* Forecasting gains depend on context quality. Left: Filtering distractors (Verified DR) consistently improves sMAE (one-sided Wilcoxon signed-rank test, significant). Right: Higher supporting-evidence coverage is positively rank-correlated with larger forecasting gains over no context.

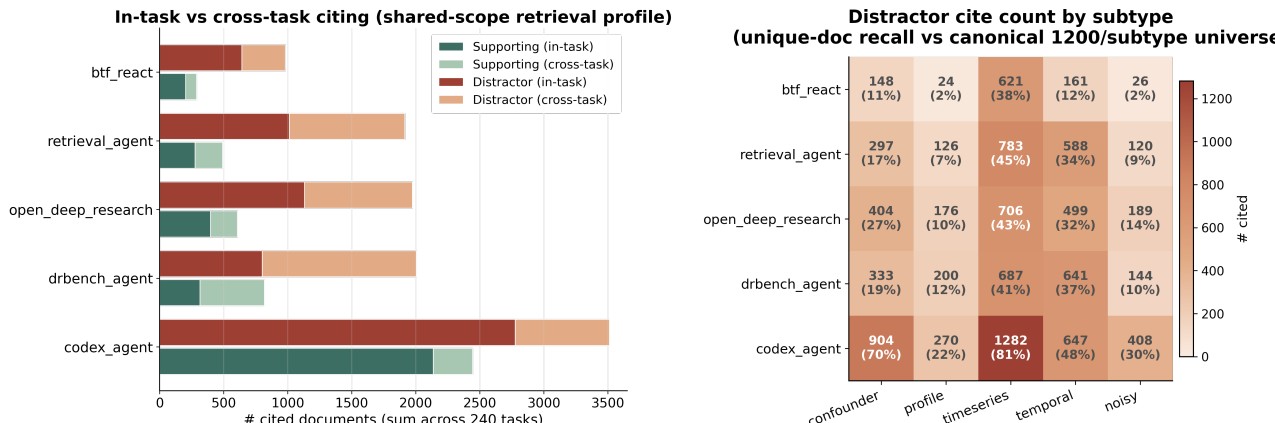

*Figure 16.* Distractor citation patterns across DR agents. Left: cited documents decomposed by whether they are supporting or distracting, and whether they come from the target task or other tasks in the shared document corpus. Right: unique cited distractor documents by taxonomy, with percentages computed against the canonical 1,200 documents per subtype. The results show two main failure modes: agents often cite off-task distractors from the shared corpus, and time-series distractors are consistently the hardest distractor type to avoid.

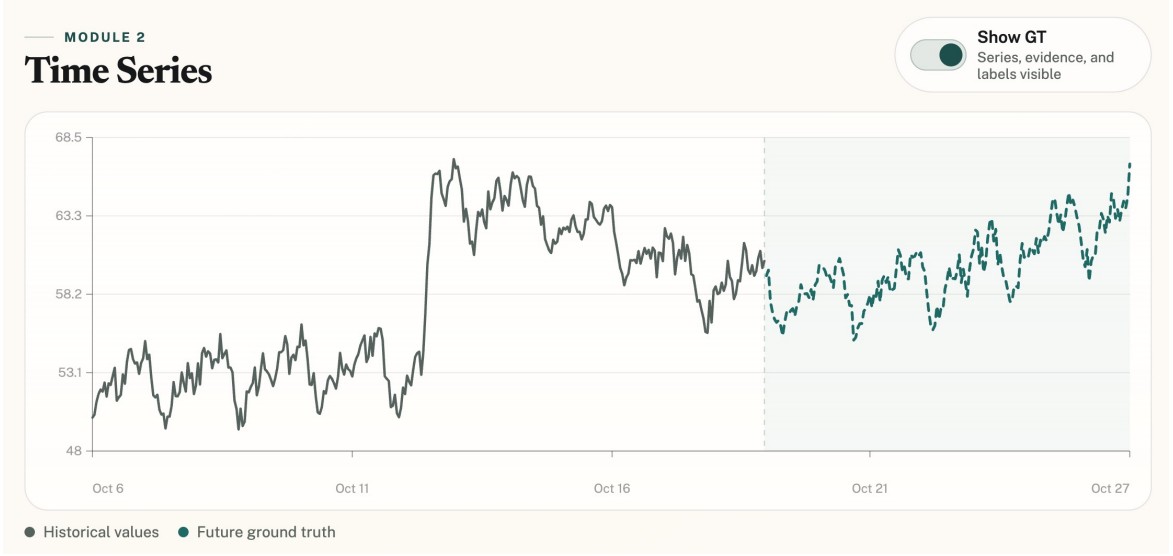

*Figure 17.* Hourly electricity usage of Riverside Cottage 12 over the historical window 2025-10-06 to 2025-10-19. The series exhibits two regimes. The first 169 readings, from 2025-10-06 00:00 to 2025-10-13 00:00, sit at a low baseline because the unit was vacant and major appliances were deactivated, the renovation work itself occupies only the final 6 hours of this window, from 2025-10-12 18:00. From 2025-10-13 01:00 onward, the household is fully occupied and the baseline jumps to a permanently elevated level that persists for the remainder of the historical window. The forecast horizon extends 180 hours past the right edge of the series.

