# OpenReview forum: "Dr-CiK: A Testbed for Foresight-Driven Agents"
_ICML.cc/2026/Workshop/FMSD — FMSD @ ICML 2026 Poster_

### Official Review · Reviewer_UH2D · 2026-05-15
**Strong benchmark for retrieval-grounded context-aided forecasting**

**Rating:** 8
**Confidence:** 3

**Review:**

This paper introduces Dr-CiK, a testbed for context-aided forecasting via deep research. In this setting, an agent is given a historical time series and a noisy document corpus, and must retrieve forecast-relevant evidence, avoid misleading distractors, synthesize useful context, and support a downstream numerical forecast. The paper contributes a benchmark with generated and expert-annotated tasks, forecast-dependent distractors, ground-truth supporting evidence, and a three-level evaluation protocol that separates end-to-end forecasting performance, deep-research retrieval quality, and controlled context-aided forecasting performance.

The main strength of the work is that it studies a highly relevant and underexplored problem.
Existing context-aided forecasting work often assumes that the useful textual context is already provided, whereas this paper examines whether agents can discover that context from a noisy corpus. This is an important setting for foundation models for structured data, since real forecasting systems often require external events, temporal grounding, causal evidence, and entity disambiguation. The benchmark design is also strong, including the distractor taxonomy is thoughtful, especially the time-series and temporal distractors, which require more than just some surface-level semantic retrieval. The three-level evaluation protocol is also useful because it allows for distinguishing failures in retrieval, evidence synthesis, distractor avoidance, and downstream forecasting.

The empirical results are interesting and support the paper’s main message. High-quality context can improve forecasting, but current deep research agents often fail to recover enough supporting evidence and frequently cite distractors, sometimes making forecasting worse than using no context. This is a useful and actionable finding. I also appreciate that the paper evaluates multiple deep-research agents and several forecasting families, including statistical baselines, pretrained time-series models, multimodal models, direct-prompt LLM forecasters, and an agentic forecaster. The appendix is unusually detailed for a workshop paper and provides useful implementation details, metrics, context conditions, prompts, and additional results.

My main concern is benchmark validity and realism. A large portion of the benchmark is generated, which is fine, but the conclusions depend on whether the generated supporting documents and distractors are sufficiently realistic proxies for real information environments. In addition, the expert-annotated task construction appears to use both historical and future windows when creating causal explanations. This is acceptable for a controlled diagnostic or pastcasting benchmark, but the paper should more clearly distinguish this from fully prospective forecasting. Relatedly, some supporting evidence appears quite close to explicit future guidance, so it would be helpful to quantify how often the supporting evidence contains direct numerical ranges, approximate target values, or strong future trajectory descriptions.

A second concern is that the comparison between deep-research agents mixes different model backends, reasoning budgets, and tool-use designs. For example, the strongest agent, i.e., Codex, appears to use a much stronger underlying model (GPT 5.5) and a much higher reasoning effort than some competing agents like Gemini 3 Flash. This is acceptable if the goal is to benchmark full systems, but the paper should be careful not to over-attribute differences to the agent architecture alone. I would also like to see main-paper results broken out by generated versus expert-annotated tasks, since the expert tasks are likely more representative of realistic deployment.

Overall, I find this to be a strong and relevant workshop submission. The benchmark is well motivated, the distractor design is thoughtful, the evaluation is diagnostic, and the empirical finding is likely to be useful for future work on forecasting agents and structured-data foundation models.

---

### Official Review · Reviewer_oM1E · 2026-05-21
**Context-Aided Forecasting Benchmark with Evaluation and Calibration Concerns**

**Rating:** 6
**Confidence:** 3

**Review:**

Summary: The paper introduces context-aided forecasting via deep research (CAF via DR), where an agent must discover forecast-relevant text from a heterogeneous corpus rather than receive it as input. To study this, the authors release Dr-CiK, a benchmark with a shared document corpus and ground truth for both the supporting evidence to be retrieved and the forecast target. A three-level protocol separates end-to-end forecasting, DR retrieval quality, and CAF under controlled context conditions. The benchmark introduces forecast-dependent distractors, generated from ground truth and history to actively mislead the forecaster, organized into a five-class taxonomy. The authors evaluate several DR agents paired with a range of forecasters and report findings on retrieval quality, distractor susceptibility, and the downstream forecasting impact of synthesized context.

Strengths:  Framing CAF as a joint retrieval-synthesis-forecasting problem is well-motivated, and providing stage-wise ground truth enables clean failure attribution. The distractor taxonomy is empirically validated by a steep hardness gradient across classes despite uniform corpus representation. Reporting hygiene is good, with multiple aggregation rules, explicit failure counts, and an internal forecaster-scaling result that directly supports the paper's claim that model size is not the binding constraint.

Weaknesses:  (W1) The central DR-agent comparison is confounded, as the strongest agent uses a different base model and a higher reasoning budget than the others, so the evidence-recall gap jointly varies architecture, base model, and reasoning budget, and the headline narrative that current DR agents are weak with one partial exception is therefore not interpretable as an architectural finding. (W2) The abstract's magnitude claim for context-driven improvement is unsupported, with within-forecaster and best-vs-best comparisons yielding meaningfully smaller ratios than the headline figure, and no aggregation rule in the paper reproduces it. (W3) There is a family-bias concern in the generation pipeline: every generation stage is gated by judges from one or two model families, and the evaluation-time recall judge is itself in the same family used during entity validation, which tightens the closed loop between what the pipeline accepts as ground truth and what is rewarded at evaluation, and the ranking is not stress-tested against an out-of-family recall judge. (W4) The expert-annotated subset, framed as the realistic and harder slice, is never broken out cleanly in the main result tables; a fine-grained difficulty decomposition partially substitutes, but a direct expert-only report is still missing, which matters because the pipeline-bias concerns apply most strongly to the synthetic majority. (W5) A smaller calibration issue: one claim describes supporting-evidence coverage as positively rank-correlated with forecasting gains, but the underlying correlation is weak rather than strong, and the main-text phrasing oversells the actual figure.

---

### Official Review · Reviewer_TgWb · 2026-05-22
**A benchmark for context-aided forecasting via deep research, with an instructive negative result the workshop audience would learn from**

**Rating:** 7
**Confidence:** 4

**Review:**

## Summary of contributions
The paper presents **Dr-CiK**, a benchmark for *context-aided forecasting via deep research* (CAF via DR). It extends the earlier **CiK** benchmark toward a more realistic, deployment-like setting. Earlier CAF benchmarks assume that useful text context is already given. Here, the agent must do more: it must find the forecast-relevant context inside a large and noisy shared corpus, reject distractors, turn the retrieved text into useful evidence, and finally produce a forecast that is grounded in this evidence. Dr-CiK provides 240 tasks (199 generated by a scalable pipeline + 41 annotated by experts) over 8,849 labeled documents in six domains. It gives ground truth for *both* retrieval and forecasting. Its main design elements are: **forecast-dependent distractors** grouped into a five-class taxonomy; anti-memorization measures (entity disambiguation and time-shifting); and a **three-level evaluation protocol** that measures retrieval quality, distractor avoidance, and context use separately. The authors evaluate five DR agents combined with classical, pretrained, multimodal, and LLM forecasters. The main result is clear: oracle context improves sCRPS strongly, but current DR agents recover only a little supporting evidence, often cite distractors, and can even make the forecast worse than using no context at all.


## Strengths
- **A problem and benchmark of clear interest to the workshop audience.** Evaluating retrieval and downstream forecasting *together*, with ground truth for both, addresses a real gap that researchers in this area would want to know about. The three-level protocol for failure attribution is a genuine methodological contribution.
- **Careful distractor design.** The forecast-dependent distractors and the five-class taxonomy test causal, temporal, entity, and numerical reasoning. The finding that time-series distractors are the main failure mode (agents cannot use the series itself as a filter) is interesting and not obvious.
- **Rigorous and well-instrumented evaluation.** The paper uses strong classical baselines (ARIMA/ETS/SES/Naive) together with pretrained, multimodal, and many LLM forecasters; it reports a probabilistic metric (sCRPS) plus calibration and sharpness; and it applies winsorization and reports failed/invalid runs explicitly.
- **Useful diagnostics.** The failure analysis shows that, even when retrieval works, the synthesis step drops numeric anchors and modal qualifiers. This is a useful insight for the community.

## Weaknesses
- **The abstract is more pessimistic than the tables.** If I understand that part correctly, "Recover < 5% (usually)" hides that the best agent (Codex/GPT-5.5) reaches 38.5% evidence recall, much higher than the others. "> 80% distractor citations" is also a small overstatement: even the *worst* agent (Retrieval, 20.4% avoidance) cites only 79.6% of distractors, and the highlighted agent (Codex, 41.0% avoidance) cites only 59%. So the real range is 59–79.6%, not always above 80%. Finally, the "nearly threefold" sCRPS reduction (abstract / §I) and the "roughly half the no-context sCRPS" number (§4.3 / Fig 4) describe *different conditions*, Fig 4 is only for DP-Gemini, while the 3× number comes from Appendix I. These should be presented as different setups, not left to look like a contradiction. The body text is more accurate and should be made consistent with the abstract.
- **No cost or latency reporting.** For an agent-based submission this is both a workshop requirement and important in practice, because the main comparisons are between agents with very different compute profiles. Interestingly, the released code already tracks this information (`drcik/utils/cost_tracking.py` records calls, tokens, `cost_usd`, and estimated cost per task/agent/model). So the data exists; it only needs to be reported in the paper.
- **The main agent comparison is confounded by the base model.** Codex uses GPT-5.5 (high reasoning effort), while the other four DR agents use Gemini-3 Flash. So "Codex is better" mixes the quality of the DR scaffold with the strength of the underlying LLM. Any claim about DR-agent *design* needs a controlled comparison that keeps the base model fixed.


## Suggestions
- Could the abstract be brought closer to the tables? For example, reporting the best-agent recall (Codex 38.5%) next to the "usually < 5%" figure, and giving the actual distractor-citation range (≈59–79.6%) rather than "> 80%", would give a more balanced picture. It would also help to note that the "nearly threefold" improvement (Appendix I) and the "roughly half" result (Fig 4, DP-Gemini) come from different conditions, so they are not read as a contradiction.
- Since the released code already records per-agent cost and tokens (cost_tracking.py), would it be possible to report these numbers in the paper? Even a small cost/latency table per DR agent and forecaster would help readers compare agents with very different compute profiles, and an iso-cost comparison would be especially informative.
- Because Codex uses GPT-5.5 while the other agents use Gemini-3 Flash, how much of Codex's advantage comes from the DR scaffold rather than the stronger base model? A small controlled comparison (for example running one other scaffold on GPT-5.5, or Codex on Gemini-3 Flash) would help separate these two factors.